# Compressed Context Memory For Online Language Model Interaction

**Jang-Hyun Kim[1,2], Junyoung Yeom[1,2], Sangdoo Yun[3]\*, Hyun Oh Song[1,2]\***
[1]Seoul National University, [2]Artificial Intelligence Institute (AIIS), [3]NAVER AI Lab
{janghyun,yeomjy,hyunoh}@mllab.snu.ac.kr, sangdoo.yun@navercorp.com

## Abstract

This paper presents a context key/value compression method for Transformer language models in online scenarios, where the context continually expands. As the context lengthens, the attention process demands increasing memory and computations, which in turn reduces the throughput of the language model. To address this challenge, we propose a compressed context memory system that continually compresses the accumulating attention key/value pairs into a compact memory space, facilitating language model inference in a limited memory space of computing environments. Our compression process involves integrating a lightweight conditional LoRA into the language model's forward pass during inference, without the need for fine-tuning the model's entire set of weights. We achieve efficient training by modeling the recursive compression process as a single parallelized forward computation. Through evaluations on conversation, personalization, and multi-task learning, we demonstrate that our approach achieves the performance level of a full context model with $5\times$ smaller context memory size. We further demonstrate the applicability of our approach in a streaming setting with an unlimited context length, outperforming the sliding window approach. Codes are available at https://github.com/snu-mllab/context-memory.

## 1 Introduction

Transformer language models have exhibited exceptional language processing capabilities, achieving remarkable results in various applications (Vaswani et al., 2017). In particular, the attention mechanism, which encompasses the entire context window, enables the language models to respond with a nuanced understanding of context. With this contextual understanding, services like ChatGPT or Bard can generate responses customized to individual users through online interactions (OpenAI, 2023; Manyika, 2023). In this online scenario, the context used for language model inference accumulates over time, raising an important challenge in efficiently handling this growing context.

A straightforward approach is to deal with previous contexts as a prompt, which leads to a continual increase in inference time and memory usage due to the growing length of contexts. Alternately, caching the attention hidden states of Transformer would be impractical (Dai et al., 2019), as the caching capacity and attention costs increase with the accumulation of contexts. Recent studies propose compressing contextual information into concise sequences of token embeddings or attention keys/values (denoted as KV) (Chevalier et al., 2023; Mu et al., 2023). However, those methods primarily focus on fixed-context scenarios and are not designed for dynamically changing contexts. Thus, they still face inefficiency and redundancy when dealing with accumulating contexts.

In this paper, we propose a novel language model framework incorporating a *compressed context memory* system for efficient online inference (Figure 1). Our memory system is capable of dynamic updates during online inference with minimal memory and computation overhead. To this end, we optimize a lightweight conditional LoRA (Hu et al., 2022), enabling language models to construct a compressed attention KV memory of contextual information through the forward computation pass. On the other hand, dynamic memory updates require a recursive context compression procedure, which leads to training inefficiencies. To address this challenge, we propose an efficient training

---
\*Corresponding authors

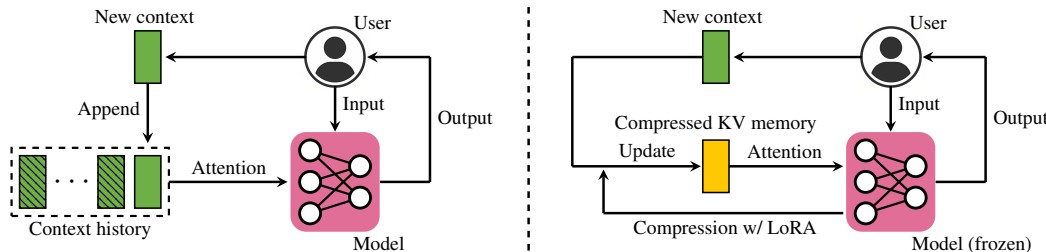

Figure 1: **Main concept of online inference systems.** Left: Conventional online inference approach. Right: The proposed system with compressed context memory. The colored boxes represent attention keys/values (or input tokens) required for Transformer inference. The *new context* refers to the sequence comprising an input and a model output from the preceding interaction.

Table 1: Analysis of inference throughput on the MetaICL dataset (Min et al., 2022) at time step 16 with LLaMA-7B and FP16 precision (Touvron et al., 2023). We measure throughput using batch processing on a single GPU. *CCM*-{concat,merge} refers to our proposed method.

|  | A100 PCIe 80GB | | | RTX 3090 24GB | | |
|---|---|---|---|---|---|---|
|  | Full context | CCM-concat | CCM-merge | Full context | CCM-concat | CCM-merge |
| Throughput (sample/sec) | 5.3 | 24.4 | **69.9** | 3.5 | 18.6 | **50.7** |
| Maximum batch size | 60 | 300 | **950** | 10 | 50 | **150** |
| Context KV length | 800 | 128 | **8** | 800 | 128 | **8** |
| Performance (Accuracy %) | **70.8** | 70.0 | 69.6 | **70.8** | 70.0 | 69.6 |

strategy that unrolls the recursive context compression procedure and processes the recursive procedure in parallel. In the inference phase, language models utilize the compressed memory to generate responses to subsequent input queries with reduced attention operations and memory.

Our approach offers several advantages compared to existing efficient context processing methods: 1) Unlike approaches that propose new attention structures such as the Linear Transformer (Katharopoulos et al., 2020), our method simply involves the integration of lightweight adapters to existing Transformer language models, leveraging the weights of pretrained models. 2) Unlike fixed-context compression techniques such as Gisting or ICAE (Mu et al., 2023; Ge et al., 2023), our approach is able to dynamically compress newly added context with minimal computational overhead. 3) In contrast to methods that recurrently compress context into token embeddings, such as RMT or AutoCompressor (Bulatov et al., 2022; Chevalier et al., 2023), our approach focuses on compressing attention keys/values, enabling a fully parallelized training process. Notably, our approach achieves a training speed that is $7\times$ faster than the mentioned approaches (Table 18) and does not require additional forward computations for the compressed token embeddings during inference.

Our online compression framework has a wide range of applications, including conversation, personalization, and multi-task learning. Notably, by compressing continuously provided dialogues, user profiles, and task demonstrations, our approach enables the language model to perform online inference with reduced memory usage and attention costs. To substantiate our claims, we evaluate our system across diverse datasets, including DailyDialog, LaMP, and MetaICL (Li et al., 2017; Salemi et al., 2023; Min et al., 2022). Through empirical analyses, we demonstrate that our method excels in both efficiency and performance compared to established context compression baselines. In particular, our method achieves equivalent performance with only $1/5$ of the context memory required when using the full context (Figure 6). This enhanced memory efficiency translates into substantial improvements in language model throughput when using batch processing on memory-constrained GPUs (Table 1). Finally, we demonstrate the efficacy of our approach in a streaming setting with an unlimited context length, outperforming the sliding window method (Figure 8).

## 2 PRELIMINARY

**Target scenario and notation** Let $\mathcal{T}$ denote a space of texts. We focus on the online inference scenario, aiming to predict the output $O(t) \in \mathcal{T}$ based on the input $I(t) \in \mathcal{T}$ and the accumulated context $C(t) = [c(1), \dots, c(t)]$ for time step $t \in [1, \dots, T]$, where $T \in \mathbb{N}$ represents

Table 2: Illustrative instances of online inference scenarios.

| Application | Dataset | Context $C(t)$ | Input $I(t)$ | Output $O(t)$ |
|---|---|---|---|---|
| Conversation | DailyDialog (Li et al., 2017) | Dialogue history | User query | Reply |
| Personalization | LaMP (Salemi et al., 2023) | User profiles | User query | Recommendation |
| Multi-task learning | MetaICL (Min et al., 2022) | Task demonstrations | Problem | Answer |

the maximum number of time steps. Here, $c(t) \in \mathcal{T}$ denotes a newly integrated context at time step $t$, which comprises the interaction results from the preceding time step $t$-1, including $I(t$-1$)$, $O(t$-1$)$, and any additional user feedback. In Table 2, we formulate diverse applications according to our target scenario and notations, where each context $C(t)$ contains accumulated information for a specific identity (*e.g.*, a task or a user). We represent the dataset with multiple identities as $\mathcal{D} = \{(C_i(t), I_i(t), O_i(t)) \mid i \in \mathcal{I}, t \in [1, \ldots, T]\}$, where $\mathcal{I}$ denotes an index set of identities. We randomly split $\mathcal{I}$ into a training set $\mathcal{I}_{\text{train}}$ and a test set $\mathcal{I}_{\text{test}}$ for experiments.

**Context compression**  Let us consider a pretrained language model $f_\theta : \mathcal{T} \to \mathbb{R}^+$, which models the probability distribution over the text space $\mathcal{T}$. A typical approach for predicting output $O(t)$ involves using the full context $C(t)$ as $\hat{O}(t) \sim f_\theta(\cdot \mid C(t), I(t))$. However, this approach requires increasing memory and computation costs over time for maintaining and processing the entire context $C(t)$. One can employ context compression techniques to mitigate this issue, compressing contexts into a shorter sequence of attention key/value pairs or soft prompts (Mu et al., 2023; Ge et al., 2023). Given the compression function $g_{\text{comp}}$, the inference with compressed contexts becomes $\hat{O}(t) \sim f_\theta(\cdot \mid g_{\text{comp}}(C(t)), I(t))$, where $|g_{\text{comp}}(C(t))| \ll |C(t)|$. It is worth noting that existing context compression methods mainly focus on compressing a fixed context $\bar{C}$ that is repeatedly used as a prompt (Mu et al., 2023; Ge et al., 2023). The objective of the compression is to generate outputs for a given input $I$ that are similar to the outputs generated when using the full context: $f_\theta(\cdot \mid g_{\text{comp}}(\bar{C}), I) \approx f_\theta(\cdot \mid \bar{C}, I)$.

## 3 METHODS

In this section, we introduce a novel approach named **Compressed Context Memory (CCM)**, designed for efficient online inference of language models. Our system compresses the given current context and dynamically updates the context memory by incorporating the compression result. We further propose a parallelized training strategy to facilitate efficient large-scale optimization.

### 3.1 COMPRESSED CONTEXT MEMORY

Here, we briefly describe the compression and inference processes at time step $t$. We denote the compressed context memory at $t$ as $\text{Mem}(t)$ with an initial value of $\text{Mem}(0) = \emptyset$. When presented with a context $c(t)$, we condense the information within $c(t)$ into the hidden feature $h(t)$ by using the compression function $g_{\text{comp}}$ as

$$h(t) = g_{\text{comp}}(\text{Mem}(t\text{-}1), c(t)). \tag{1}$$

The compressed context memory $\text{Mem}(t)$ is then updated via an update function $g_{\text{update}}$ as

$$\text{Mem}(t) = g_{\text{update}}(\text{Mem}(t\text{-}1), h(t)). \tag{2}$$

Within a limited memory space, $\text{Mem}(t)$ stores contextual information up to time $t$. By encompassing only the input $I(t)$ and memory $\text{Mem}(t)$, we conduct memory-efficient inference as

$$\hat{O}(t) \sim f_\theta(\cdot \mid \text{Mem}(t), I(t)). \tag{3}$$

In the following, we elaborate on the compression and update processes.

**Compression**  We compress context information into attention keys/values as in Compressive Transformer (Rae et al., 2020) and Gisting (Mu et al., 2023). This compression approach can be applied within each layer of the language model, providing better parallelization than the auto-encoding approach (Ge et al., 2023). We introduce a specialized compression token $\langle\texttt{COMP}\rangle$ and

train the language model to compress context information into the attention keys/values of the $\langle\text{COMP}\rangle$ token, similar to the Gisting approach.

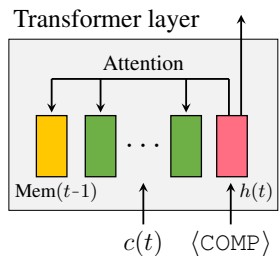

Transformer layer

We assume a Transformer language model $f_\theta$ has $L$ layers with a hidden state dimension of $d$. To simplify notation, we set a compression token length of 1. It is worth noting that the compression token can be extended to arbitrary lengths. Under these conditions, the total size of the attention keys/values of $\langle\text{COMP}\rangle$ token is $2 \times L \times d$. The compression process is illustrated in Figure 2. At each time step $t$, we append $\langle\text{COMP}\rangle$ token to the context $c(t)$ and make the $\langle\text{COMP}\rangle$ token to have attention on the keys/values of $c(t)$ and the previous memory state Mem$(t\text{-}1)$. Utilizing the resulting attention keys/values of the $\langle\text{COMP}\rangle$ token, we obtain the compressed hidden feature $h(t) \in \mathbb{R}^{2 \times L \times d}$ in Equation (1).

Figure 2: The illustration of the compression process at time step $t$. Each colored box symbolizes attention hidden states.

**Memory update**  We propose memory update functions $g_{\text{update}}$ that are differentiable and parallelizable during training. In particular, we consider the simplest form of $g_{\text{update}}$ and verify the effectiveness of our compression framework. Considering various application scenarios, we examine two types of memory systems: 1) a scalable memory and 2) a fixed-size memory, similar to an RNN.

- For a scalable memory setting, we employ the *concatenation* function as $g_{\text{update}}$. Then Mem$(t) \in \mathbb{R}^{t \times 2 \times L \times d}$ contains the attention key/value pairs associated with $\langle\text{COMP}\rangle$ tokens up to time step $t$. We denote our system with the concatenation function as ***CCM-concat***.

- For a fixed-size memory system, we propose a *merging* function to update information in the memory. Specifically, we update memory by weighted average: Mem$(t) \in \mathbb{R}^{2 \times L \times d}$ as Mem$(t) = (1-a_t)\text{Mem}(t\text{-}1) + a_t h(t)$, where $a_1 = 1$ and $a_t \in [0, 1]$ for $t \geq 2$. With this recurrence, Mem$(t)$ becomes Mem$(t) = \sum_{j=1}^{t} a_j \prod_{k=j+1}^{t}(1 - a_k)\ h(j)$. In the main experiments, we evaluate an update method based on the arithmetic average of the compressed states with $a_t = 1/t$, *i.e.*, Mem$(t) = \frac{1}{t}\sum_{j=1}^{t} h(j)$. We denote our method with the merging function as ***CCM-merge***.

During training, we compute Mem$(1), \ldots,$ Mem$(t)$ in parallel by averaging hidden features $h(1), \ldots, h(t)$ simultaneously. In the online inference phase, we recurrently update the memory by cumulative average using the prior memory Mem$(t\text{-}1)$ and current compression result $h(t)$. In Appendix Table 14, we examine another design choice for the merge function: the exponential moving average. It is also worth noting that CCM-concat can be interpreted as a process that dynamically infers coefficients for hidden states $h(t)$ through the attention mechanism.

**Parallelized training**  The direct integration of the compression process of Equation (1) into the training process poses a challenge as it requires recursive model executions over $j = 1, \ldots, t$. Such recursive executions prolong training time and amplify back-propagation errors through the elongated computation graph (Gruslys et al., 2016). To overcome this challenge, we propose a fully parallelizable training strategy, taking advantage of the Transformer structure.

For training data $(C(t), I(t), O(t)) \in \mathcal{D}_{\text{train}}$, we insert $\langle\text{COMP}\rangle$ tokens into the accumulated context $C(t)$, forming the sequence $[c(1), \langle\text{COMP}\rangle \cdots c(t), \langle\text{COMP}\rangle, I(t)]$. We then establish memory update and attention mechanisms, modeling recursive compression processes as parallelized forward computations (Figure 3). In detail, within each layer of a Transformer $f_\theta$, we update Mem$(j)$ for $j \leq t$ using the attention keys/values of preceding $\langle\text{COMP}\rangle$ tokens, *i.e.*, $h(1), \ldots, h(j)$, as in Figure 3 (a). Following the memory update, we execute the compression procedures for $j = 1, \ldots, t$ in parallel using the masked attention as in Figure 3 (b). As stated in Equation (3), we access the context information from previous time steps only through memory during online inference. Therefore, we restrict $c(j)$ to reference only Mem$(j\text{-}1)$ for $j \leq t$ and make $I(t)$ exclusively have its attention on Mem$(t)$. Finally, we compute likelihood $f_\theta(O(t) \mid \text{Mem}(t), I(t))$ in Equation (3) using the output probability obtained at the last token position of $I(t)$. When the token length of $O(t)$ exceeds 1, we follow the conventional approach by conditioning on the target label $O(t)$ and calculating the loss for the next tokens (Radford et al., 2019). All these steps take place within a single forward pass of $f_\theta$, and the loss gradients are backpropagated to all tokens across all time steps.

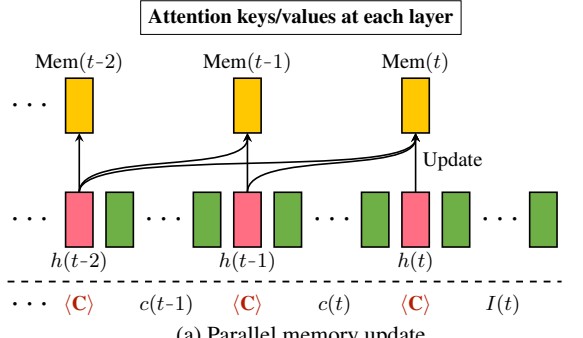
(a) Parallel memory update

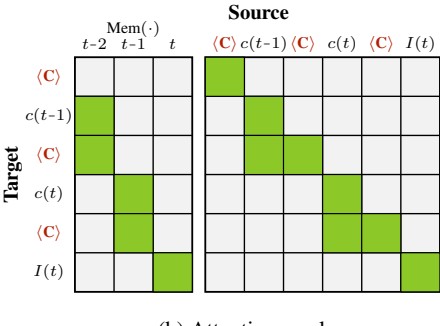
(b) Attention mask

Figure 3: **Illustration of the parallelized training process.** In (a), each colored box symbolizes attention keys/values of memory, compression tokens, and normal text tokens. In (b), gray indicates that attention is blocked. In the figures, ⟨**C**⟩ stands for ⟨COMP⟩. At each layer, after the parallel updates of compressed context memory, the attention operation occurs with the mask in (b). Note the calculation of Mem($t$) occurs after $c(t)$ and its subsequent ⟨COMP⟩ token. Reordering the top row of (b) to align with this temporal relation yields an autoregressive mask.

---

**Algorithm 1** Training stage for compression

---

**Input:** Language model $f_\theta$, training set $\mathcal{D}_\text{train}$
Initialize a conditional LoRA weight $\Delta\theta$
Modify the forward pass of $f_\theta$ to update the compressed context memory
**repeat**
    Sample a mini-batch $\mathcal{B} \subset \mathcal{D}_\text{train}$ and set $\mathcal{B}' = \emptyset$
    **for** $(C_i(t), I_i(t), O_i(t)) \in \mathcal{B}$ **do**
        Prepare an input $x_i = [c_i(1), \langle\text{COMP}\rangle, \ldots, c_i(t), \langle\text{COMP}\rangle, I_i(t)]$ and a target $y_i = O_i(t)$
        $\mathcal{B}' = \mathcal{B}' \cup \{(x_i, y_i)\}$
    **end for**
    Compute loss in eq. (4) on $\mathcal{B}'$ through a single forward pass using the masked attention
    Perform a gradient descent step *w.r.t.* $\Delta\theta$
**until** convergence
**Output:** $\Delta\theta$

---

**Conditional adapter** Current compression methods typically rely on fine-tuning a language model $f_\theta$ to acquire compression capabilities (Mu et al., 2023). In this approach, the construction of the memory hinges on the adjustment of the language model parameter $\theta$, allowing us to parameterize the memory for context $C_i(t)$ as $\text{Mem}_i(t; \theta)$. The objective function for learning compression capability is then formulated as $\min_\theta \mathbb{E}_{t, i \sim \mathcal{I}_\text{train}}[-\log f_\theta(O_i(t) \mid \text{Mem}_i(t; \theta), I_i(t))]$.

However, this conventional objective can potentially lead the language model to generate answers for input $I_i(t)$ without considering the memory $\text{Mem}_i(t; \theta)$. Such overfitting to the input $I_i(t)$ can diminish the importance of compressed context memory during training, which leads to insufficient training of the compression capability. Specifically, when we measure the loss without context, $\mathbb{E}_{t, i \sim \mathcal{I}}[-\log f_\theta(O_i(t) \mid I_i(t))]$, throughout the compression training process with LLaMA-7B on MetaICL, the loss on training set decreases from 2.69 to 1.84, whereas the loss on test set remains 2.59. This observation indicates the presence of overfitting on inputs.

To address this issue, we introduce separate trainable parameters specifically for compression. To this end, we propose a conditional variant of LoRA (Hu et al., 2022), which operates exclusively on ⟨COMP⟩ tokens. This ensures that the trainable parameters allocated for compression solely influence the model's compression capabilities (Figure 4). Let $W \in \mathbb{R}^{d \times d}$ denote a parameter of a feed-forward layer with a hidden dimension $d$, and let $\Delta W = A^\intercal B \in \mathbb{R}^{d \times d}$ denote a corresponding LoRA weight with $A, B \in R^{k \times d}$ and $k \ll d$. For input token $x$ and its corresponding hidden

Figure 4: Feed forward operations of our conditional LoRA.

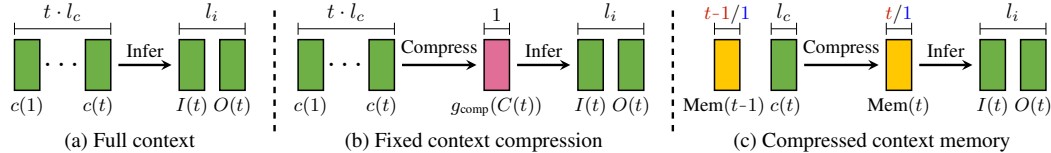

Figure 5: **Illustration of the compression and inference processes** at time step $t$. The arrow indicates the process of referencing the keys/values on the left to generate the keys/values on the right. Here, $l_c$ means the expected length of key/value pairs of context $c(\cdot)$, and $l_i$ denotes the total length of input and output. We assume that each compression outcome has a length of 1. Notations at the top of $\text{Mem}(\cdot)$ denote the length of key/value pairs corresponding to CCM-concat/-merge.

Table 3: Complexity analysis of approaches in online inference scenario at time step $t$. Figure 5 presents illustrative explanations for the compression/inference processes with respective notations.

| Type | Operation | Full context | Fixed-context compression | CCM-concat | CCM-merge |
|------|-----------|--------------|---------------------------|------------|-----------|
| Memory | Compression | - | $O(tl_c)$ | $O(t + l_c)$ | $O(l_c)$ |
|  | Inference | $O(tl_c + l_i)$ | $O(l_i)$ | $O(t + l_i)$ | $O(l_i)$ |
| Attention | Compression | - | $O(tl_c)$ | $O(t + l_c)$ | $O(l_c)$ |
| FLOPS | Inference | $O(tl_cl_i + l_i^2)$ | $O(l_i^2)$ | $O(tl_i + l_i^2)$ | $O(l_i^2)$ |

state $x_h \in \mathbb{R}^d$, we propose the following conditional forward computation:

$$x_h' = Wx_h + m \cdot \Delta W x_h,$$

where $m = \mathbb{1}(x = \langle\text{COMP}\rangle)$. We denote all trainable LoRA parameters of a model as $\Delta\theta$. The parameter $\Delta\theta$ only affects the formation of compressed context memory, and our compression training objective with conditional LoRA is

$$\underset{\Delta\theta}{\text{minimize}} \ \mathbb{E}_{t,i\sim\mathcal{I}_{\text{train}}} \left[ -\log f_\theta(O_i(t) \mid \text{Mem}_i(t; \theta + \Delta\theta), I_i(t)) \right]. \quad (4)$$

We summarize the training procedure of our approach in Algorithm 1.

## 3.2 COMPLEXITY ANALYSIS

We analyze the complexity of approaches in online inference scenarios in Table 3. In the table, "full context" refers to the method using full context $C(t)$ during inference, and "fixed-context compression" refers to the method compressing $C(t)$ as $g_{\text{comp}}(C(t))$ at each time step (Mu et al., 2023). In Figure 5, we visualize these methods and introduce notations used in complexity analysis.

Regarding the full context method, the context length at time step $t$ is $tl_c$, resulting in inference memory complexity of $O(tl_c + l_i)$ and quadratic attention FLOPS of $O(tl_cl_i + l_i^2)$. Fixed-context compression methods offer reduced complexity for inference. However, they process the entire context $C(t)$ for compression, resulting in memory and FLOPS complexities of $O(tl_c)$.

Our method, utilizing compressed context memory for both compression and inference, exhibits reduced complexity. In the case of CCM-merge, compression complexity depends solely on the length of context $c(t)$ as $O(l_c)$. For CCM-concat, the complexity becomes proportional to the time step $t$ due to growing memory size over time. Nonetheless, the compression complexity reduces from $O(tl_c)$ to $O(t + l_c)$ when compared to fixed-context compression methods. While CCM-concat exhibits higher complexity than CCM-merge, a language model using CCM-concat achieves superior performance, offering a trade-off between performance and complexity (Figure 6).

## 4 EXPERIMENTS

In this section, we present the empirical validation of our approach in online inference scenario. In Section 4.2, we further substantiate our claims through an ablation study and additional analyses. We provide a detailed training setup in Appendix D.

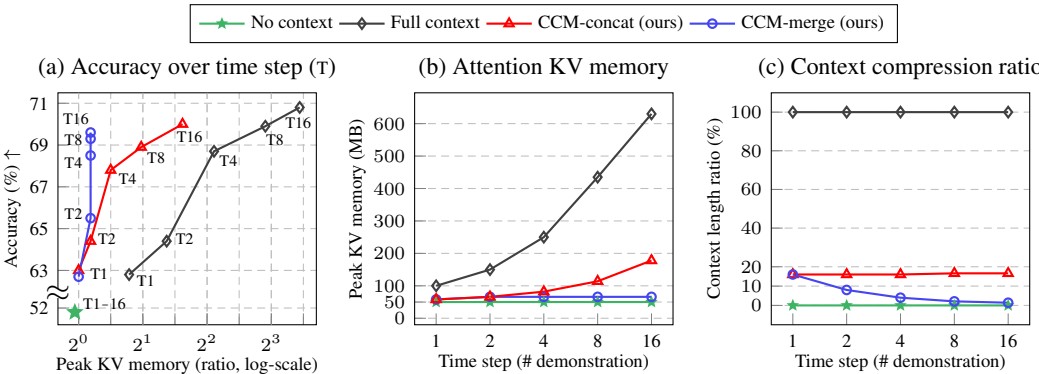

Figure 6: Comparison to full context approach on MetaICL test tasks with LLaMA-7B. *Peak KV memory* refers to the peak memory space occupied by attention keys/values during compression and inference processes at each time step. We provide results for other datasets in Appendix, Figure 10.

**Datasets and metrics**    We conduct evaluations using three datasets: MetaICL (Min et al., 2022), LaMP (Salemi et al., 2023), and DailyDialog (Li et al., 2017). First, MetaICL is a dataset for multi-task in-context learning, aiming at solving tasks unseen during training. We evaluate on the high-to-low resources setting, consisting of 61 training tasks and 26 unseen test tasks. The evaluation metric is accuracy for multiple-choice questions. Next, LaMP is a dataset for personalization, utilizing user profiles to generate personalized recommendations. For evaluation, we measure the accuracy of multi-choice recommendations on new users unseen during training. Lastly, we assess performance in conversation scenarios using the DailyDialog dataset, comprising sequences of everyday conversations. We evaluate models by measuring perplexity on actual dialogues. In Appendix C, we provide more detailed information and statistics for each dataset.

**Baselines**    We implement established fixed-context compression techniques with open-source codes. Our primary focus is on evaluating the *Compressive Transformer* (Rae et al., 2020) and *Gisting* (Mu et al., 2023), both designed to compress attention hidden states. To suit online inference scenarios, we devise Gisting to compress contexts $c(1), \ldots, c(t)$ separately and evaluate the method using the concatenated compression results for inference. We refer to this approach as *Gisting-online*. For the recurrent compression approaches, *RMT* and *AutoCompressor* (Bulatov et al., 2022; Chevalier et al., 2023), we conduct a separate comparison as publicly available trained models are limited to the OPT architecture (Zhang et al., 2022). We also evaluate the performance of language models using *full context* to quantify the performance degradation due to compression.

## 4.1 COMPRESSION PERFORMANCE

**Comparison to full context method**    In Figure 6, we analyze the memory efficiency of our method in an online inference scenario. Figure 6-a shows the performance obtained at each time step, along with the peak memory required for attention keys/values during the compression and inference processes illustrated in Figure 5. The results demonstrate the memory efficiency advantage of our approach compared to the full context approach. Specifically, CCM-concat achieves comparable performance by using half the key/value memory space, whereas CCM-merge attains equivalent performance levels with approximately $1/8$ of the key/value memory space. While CCM-concat requires more memory, it outperforms the merge approach as time steps increase. Compared to the *No context* method, which relies solely on inputs to generate outputs, our methods exhibit superior performance with a negligible increment in context memory size. Remarkably, our method demonstrates an 18% boost in performance compared to the no-context method at time step 16.

**Comparison to compression baselines**    Figure 7 compares the test performance of compression methods on various datasets. For a fair comparison, we set an identical compression factor for all compression methods, except for CCM-merge, which has a higher compression factor. The figure shows that our compressed context memory approach consistently outperforms established compression baselines across all time steps, demonstrating performance that closely parallels the full context approach. Regarding the Gisting approach, which is optimized for compressing a fixed

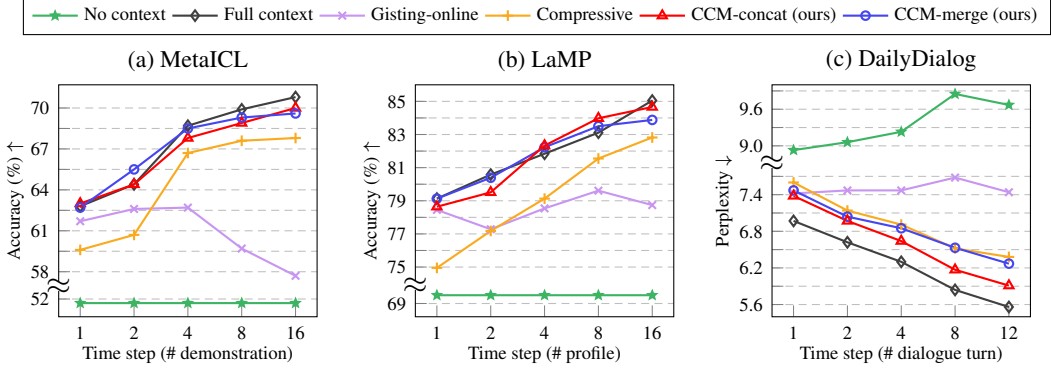

Figure 7: Test performance of compression methods in online inference scenario with LLaMA-7B. All compression methods have the **identical compression factor** around 8, except for CCM-merge, which has a higher compression factor. We provide exact values in Appendix, Tables 23 to 25.

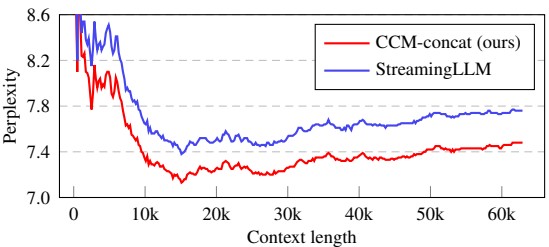

Figure 8: Streaming evaluation on PG19 validation set using sliding window with LLaMA-7B.

Figure 9: KV cache during streaming with CCM-concat. The example above assumes a CCM maximum size of 4 and a sliding window maximum size of 8.

context in a single iteration, there is no performance improvement as the time step increases. It is worth noting that there is a key distinction among the datasets considered. Regarding MetaICL, the task demonstrations $c_i(1), \ldots, c_i(t)$ are mutually complementary, sharing information related to the $i^{\text{th}}$ task. Similarly, LaMP's user profiles share information about specific users. On these datasets, both merge and concatenation approaches yield similar performance, indicating insignificant compression loss during the merge operation. On the other hand, in the dialogue dataset, the contexts $c_i(1), \ldots, c_i(t)$ conveyed through the $i^{\text{th}}$ conversation have distinct information. In this case, the concatenation approach, which compresses context information into distinct memory spaces, outperforms the merge approach as shown in Figure 7-c. This observation indicates that as diverse information is introduced over time, the loss of information in the merge approach increases.

**Unified compression adapter**   To demonstrate the generalization ability of our method in more general scenarios, we train a single compression model and evaluate its performance across various tasks. Specifically, we leverage MetaICL training tasks and a conversation dataset, SODA (Kim et al., 2023), as our training data, and then evaluate on multiple test tasks: MetaICL unseen test tasks, LaMP, and DailyDialog. In Appendix E, Table 10, we provide evaluation results of the single compression model. We note that the compression performance decreases slightly compared to a compression adapter trained specifically for each application (Figure 7). For example, on the MetaICL test tasks, the compression accuracy gap increases from 0.8% to 1.3%. However, Table 10 shows that our method obtains the best compression ability across all evaluation sets, demonstrating our approach's generalization ability on data and scenarios unseen during training.

**Streaming with sliding window**   We incorporate CCM into the sliding window approach with attention sink (Xiao et al., 2023). During streaming, tokens are processed one by one while adhering to the limited KV cache memory size. When the KV cache limit is reached, we compress the oldest tokens in the context window to update the compressed memory (Figure 9). In the case of CCM-concat, we manage the compressed memory size by emitting the oldest compressed key/value pair. Following Xiao et al. (2023), we reassign sequential position IDs starting from 0 within the KV

Table 5: An example result using our method with LLaMA-7B on a DailyDialog test sample.

| |
|---|
| **Context**: |
| **A**: What's the problem, Nada? You look down in the dumps. ⟨COMP⟩ |
| **B**: I don't know. My life is a big mess. Everything is so complicated. ⟨COMP⟩ |
| **A**: Come on, nothing can be that bad. ⟨COMP⟩ |
| **B**: But promise me, you'll keep it a secret. ⟨COMP⟩ |
| **A**: Ok, I promise. So what's troubling you so much? ⟨COMP⟩ |
| **B**: I've fallen in love with my boss. ⟨COMP⟩ |
| **A**: Really? Is he married? ⟨COMP⟩ |
| ⟹ Total **103** tokens. Context compression ratios are **7/103** (CCM-concat) and **1/103** (CCM-merge). |
| **Input**: No, of course not. He is still single. |
| **Output generated w/o context**: I'm sorry, I'm not sure what you mean. |
| **Output generated by CCM-concat**: So what's the problem? |
| **Output generated by CCM-merge**: What's his problem? |
| **Ground truth output**: Then what's your problem? |

cache in every streaming step. In Figure 8, we compare our approach to StreamingLLM (Xiao et al., 2023), which only stores the most recent keys/values in the sliding window. To ensure a fair comparison, we modify the baseline method to have an identical KV cache size as our approach at every streaming step. We use the Pretrain+MetaICL+SODA 500k model in Table 11, and conduct evaluation on the PG19 validation set (Rae et al., 2020). Specifically, we set the maximum KV size to 160 and the CCM size to 8, while compressing 64 tokens to a size of 2 at each compression step. The results in Figure 8 demonstrate the effectiveness of our compression method in the streaming setting, outperforming the StreamingLLM approach.

## 4.2 ANALYSIS

In this section, we provide quantitative and qualitative analyses of our method. In Appendix E, we provide supplementary experimental results on **training data sources, compression FLOPS, compression token length, larger model scales, and comparison to other baselines**, including AutoCompressor and MemoryBank (Chevalier et al., 2023; Zhong et al., 2023).

**Effect of conditional LoRA**  In Equation (4), we compare the compression performance of our conditional LoRA with the default unconditional LoRA. Table 4 shows evaluation results obtained using the identical training recipe. The table confirms the consistent superiority of our conditional LoRA over the default unconditional LoRA across all methods, including Gisting. In Appendix Table 21, we provide results on LaMP and DailyDialog, demonstrating that our conditional LoRA consistently improves the performance.

Table 4: Test accuracy (%) of default LoRA and our conditional LoRA with LLaMA-7B on MetaICL at time step 16.

| Method | Default | Conditional (ours) |
|---|---|---|
| CCM-concat | 69.4 | **70.0** (+0.6) |
| CCM-merge | 66.3 | **69.6** (+3.3) |
| Gisting | 64.6 | **66.9** (+2.3) |

**Qualitative results**  Table 5 illustrates the results of applying our approach to DailyDialog, using a ⟨COMP⟩ token length of 1. The table shows that our methods continue a seamless conversation within the given context, while CCM-concat generates a response that better suits the overall context.

## 5 CONCLUSION

We present a novel compressed context memory system that dynamically compresses contextual information, thereby enhancing the online inference efficiency of language models. To ensure efficient training, we develop a parallelized training strategy and introduce a conditional adapter. Our approach achieves reduced memory and attention FLOPS complexities compared to previous fixed-context compression methods. We validate the practical applicability of our approach through a comprehensive evaluation on multi-task learning, personalization, and conversation applications.

ACKNOWLEDGEMENT

This work was supported by SNU-NAVER Hyperscale AI Center, Institute of Information & Communications Technology Planning & Evaluation (IITP) grant funded by the Korea government (MSIT) [No. 2020-0-00882, (SW STAR LAB) Development of deployable learning intelligence via self-sustainable and trustworthy machine learning and NO. 2021-0-01343, Artificial Intelligence Graduate School Program (Seoul National University)], and Basic Science Research Program through the National Research Foundation of Korea (NRF) funded by the Ministry of Education (RS-2023-00274280). Hyun Oh Song and Sangdoo Yun are the corresponding authors.

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

APPENDIX

## A    RELATED WORKS

**Context compression**    Seminal works, such as Memory Networks, have introduced novel models and computational approaches to efficiently store contextual information within limited space, enhancing the inference efficiency of language models (Weston et al., 2015; Ba et al., 2016). Recently, there have been efforts to compress frequently used prompts, aiming to enhance the inference efficiency of large-scale language models. Wingate et al. (2022) advocate condensing prompts into concise soft prompts. Hyper-Tuning (Phang et al., 2023) attempts to convert prompts into model adapters, while Snell et al. (2022) propose distilling prompt information into the model parameters. AutoCompressor (Chevalier et al., 2023) and ICAE (Ge et al., 2023) propose auto-encoding approaches for compressing contexts into soft embeddings. Gisting (Mu et al., 2023) introduces learnable tokens designed to compress context information within attention hidden states. These previous methods focus on compressing fixed context to enhance reusability. In this study, we introduce a task involving context compression during online inference and propose an effective approach for handling dynamically changing contexts.

**Long context Transformer**    In terms of efficient context processing, our approach relates to the long context Transformer. Notably, Dai et al. (2019) aims to increase the context length through attention hidden state caching, and Rae et al. (2020) proposes a strategy to compress attention hidden states. Efforts have also focused on reducing the complexity of attention operations (Child et al., 2019; Zaheer et al., 2020). These methods, which propose new attention mechanisms, require training large models from scratch, making it challenging to leverage existing pretrained models. The following works propose recurrent memory-augmented approaches (Bulatov et al., 2022; Hutchins et al., 2022), while Wu et al. (2022) propose k-nearest retrieval of attention key/value pairs to manage long contexts. These retrieval-based approaches, including MemoryBank (Zhong et al., 2023) and LongMem (Wang et al., 2023), primarily focus on the token-level retrieval process, with less emphasis on memory compression. However, LLM's keys and values demand a significant amount of storage, reaching several hundred megabytes even for a context length of 1024. Such high storage requirements can become problematic in scenarios such as user-level personalization and conversation systems. Recently, notable attempts have been made to extend the context length of LLaMA (Mohtashami & Jaggi, 2023; Tworkowski et al., 2023). While these studies concentrate on handling fixed contexts, our approach aims to dynamically compress expanding contextual information within a compact memory space.

## B    DISCUSSIONS

**Application-specific compression**    When focusing on specific applications, the size of compressible contextual information becomes larger than when considering general scenarios (Tishby & Zaslavsky, 2015). This indicates that application-specific compression modules can achieve higher compression efficiency compared to their more general counterparts. Similar to fine-tuning foundation models for specific applications in various industries, an application-specific compression module can be employed to achieve superior compression capability. It is noteworthy that our method is application-agnostic, meaning it can be applied effectively to a wide range of scenarios in a data-driven manner. Obtaining a compression module without requiring application-specific knowledge or manual adjustments holds practical value. Furthermore, as demonstrated in the generalization test presented in Table 10, our approach shows generalization capabilities across various applications and can be flexibly adapted to different scenarios.

**Limitations and future works**    While our model is capable of generalizing to new tasks or user contexts at test time, training a broadly applicable model for arbitrary applications remains an important future direction. Moreover, despite surpassing existing compression baselines in performance, our approach still declines in performance compared to when utilizing the full context. Developing compression techniques that can ensure a higher level of information preservation remains a crucial direction for future research.

## C  DATASET DETAILS

Table 6 illustrates training data formats. While all datasets adhere to the same format, the content within each context and input varies. In Table 7, we provide statistics for each dataset. In the case of MetaICL, we employ the *high-to-low-resource* setting consisting of a total of 61 training tasks and 26 test tasks, which is the most representative setting (Min et al., 2022). The token length of demonstrations varies depending on the dataset type. We filter out demonstrations exceeding a token length of 256 in both the training and evaluation sets. Taking into account the GPU memory constraints, we set the maximum token length for the entire context to 1024. For LaMP, we conduct evaluations in the *personalized categorization* setting (Salemi et al., 2023). The dataset exhibits relatively lower token length variations than MetaICL. Regarding DailyDialog, as more than 90% of the test samples have dialogue turns of 12 or fewer, we set the maximum time step to 12.

Table 6: Illustrative format of each dataset sample.

| Dataset | Context $C_i(t)$ with $\langle$COMP$\rangle$ token | Input $I_i(t)$ |
|---|---|---|
| MetaICL | Demonstration 1 for task $i$ $\langle$COMP$\rangle$ $\cdots$ Demonstration $t$ for task $i$ $\langle$COMP$\rangle$ | A problem for task $i$ |
| LaMP | Profile 1 for user $i$ $\langle$COMP$\rangle$ $\cdots$ Profile $t$ for user $i$ $\langle$COMP$\rangle$ | A query for user $i$ |
| DailyDialog | Turn 1 from dialog $i$ $\langle$COMP$\rangle$ $\cdots$ Turn $t$ from dialog $i$ $\langle$COMP$\rangle$ | Turn $t$+1 from dialog $i$ |

Table 7: Descriptions for datasets considered.

| | MetaICL | LaMP | DailyDialog |
|---|---|---|---|
| Average token length of context $c(\cdot)$ at each time step | 50 | 50 | 15 |
| Maximum token length of context $c(\cdot)$ at each time step | 256 | 100 | 128 |
| Maximum time step $T$ | 16 | 16 | 12 |

## D  EXPERIMENT SETUP

**Training setup**  We begin by fine-tuning LLaMA pretrained models (Touvron et al., 2023) on each training dataset. The performance of these models with full contexts establishes the upper-bound performance of our experiment. We then perform LoRA fine-tuning on these models to learn compression capabilities. To ensure a fair comparison, we employ identical LoRA configurations and training protocols across all methods considered. All experiments undergo training with a fixed number of data, ranging from 10k to 250k, depending on the datasets. Individual training runs take 3 to 24 hours on a single NVIDIA A100 with 80GB memory. To account for limited GPU memory, we set the maximum token length of each training sample to 1024. Regarding Gisting, utilizing our conditional adapter enhances performance (Table 4). Based on this observation, we report the improved performance achieved by applying our conditional adapter in the main experiment. To confirm the effectiveness of compression, we adjust the length of $\langle$COMP$\rangle$ tokens to attain a sufficiently large **compression factor of approximately 8** for each dataset. For specific training recipes and hyperparameters, please refer to Appendix D.

**Training protocol and hyperparameter**  Our approach first fine-tunes the pretrained LLaMA models on each training dataset, following the training recipe in Table 8 and the LORA configuration in Table 9. The resulting LORA adapters are then merged with the pre-existing model weights. Using this fine-tuned model as a foundation, we proceed to train the compression capability. To ensure a fair comparison, we optimize both compression baselines and our methods using the same training recipe in Table 8 and LORA configuration in Table 9. We jointly optimize the embeddings for $\langle$COMP$\rangle$ tokens, where $\langle$COMP$\rangle$ tokens at different time steps share the same embedding. All training processes are conducted on a single A100 PCIe 80GB GPU and take 3 to 24 hours, depending on the dataset.

**Evaluation method**  For MetaICL and LaMP, we measure the accuracy for multi-choice questions by comparing the average log-likelihood on tokens of each answer choice, following the official evaluation codes provided by MetaICL (Min et al., 2022).

Table 8: Training recipes of our experiments for LLaMA models.

|  | MetaICL | LaMP | DailyDialog |
|---|---|---|---|
| Training steps | 2000 | 300 | 1000 |
| Batch size | 128 | 128 | 128 |
| # training samples | 256k | 38k | 128k |
| Learning rate | 3e-4 | 3e-4 | 3e-4 |
| Learning rate scheduling | Cosine | Cosine | Cosine |
| Mixed precision | FP16 | FP16 | FP16 |
| ⟨COMP⟩ token length | 8 | 4 | 2 |

Table 9: LoRA configurations for LLaMA models. We use this configuration for all experiments.

| Argument | Setting |
|---|---|
| Target modules | q_proj,k_proj,v_proj,o_proj |
| Rank | 8 |
| Alpha | 16 |
| Dropout | 0.05 |

# E   ADDITIONAL EXPERIMENTAL RESULTS

## E.1   MAIN ANALYSIS

**Unified compression adapter**   We train a single compression adapter with LLaMA-7B on the mixture of the MetaICL training tasks and the SODA conversation dataset. We follow the training recipe in Table 8, while we train a model for 4k steps. We train the Gisting and Compressive Transformer baselines using the same dataset and training protocol. We use the ⟨COMP⟩ token length of 2 for CCM-concat and 8 for CCM-merge. Finally, we test the model on the MetaICL unseen test tasks, LaMP, and DailyDialog at the corresponding maximum time step.

Table 10 demonstrates the generalization ability of our approach on datasets and scenarios unseen during training. Specifically, CCM-concat maintains the best compression performance by a large margin compared to baseline methods. We observe that CCM-merge has increased performance degradation by compression compared to the scenario-specific settings (*e.g.*, the LaMP accuracy degradation by compression increased from 1.2% to 5.1%). However, the other compression baselines have a larger performance gap by compression, demonstrating our approach achieves the best generalization performance among the baselines.

Table 10: Evaluation of a single model trained on MetaICL and SODA training datasets. *Memory* refers to the peak memory required for attention keys/values during inference.

| Test dataset | Metric | No context | Full context | Gisting-online | Compressive | CCM-concat | CCM-merge |
|---|---|---|---|---|---|---|---|
| MetaICL | Accuracy (%) | 53.6 | 70.0 | 59.9 | 65.0 | **68.7** | 67.8 |
|  | Memory (MB) | 50 | 630 | 82 | 82 | 82 | 66 |
| LaMP | Accuracy (%) | 37.0 | 76.4 | 67.6 | 58.4 | **75.2** | 71.4 |
|  | Memory (MB) | 50 | 755 | 82 | 82 | 82 | 66 |
| DailyDialog | Perplexity | 11.51 | 7.02 | 9.04 | 9.19 | **7.61** | 8.22 |
|  | Memory (MB) | 32 | 252 | 54 | 54 | 54 | 38 |

**Effect of training data sources**   To analyze the impact of data used for compression adapter training, we compare the performance of CCM-concat trained with various data sources. Table 11 presents evaluation results using RedPajama-V2 (Computer, 2023) and LmSys-Chat (Zheng et al., 2023) as the base training data. The table shows that the evaluation performance improves when using training data from similar sources. Particularly, when adding a new data source, the performance in the added data source significantly improves with a marginal performance decrease in the existing data sources. We believe that different data sources have different compressible information spaces, indicating the importance of constructing training data tailored to the application scenario. Lastly, it

is worth noting that increasing the amount of training data enhances overall performance (last row in Table 11), underscoring the significance of both the quantity and quality of the training data.

Table 11: Compression performance gap across different data sources used to train compression adapter. We measure the perplexity gap compared to the full context method at the maximum time step (accuracy for MetaICL). We use CCM-concat with ⟨COMP⟩ token length of 2 on LLaMA-7B.

| Training dataset | # training data | Evaluation dataset | | | |
|---|---|---|---|---|---|
| | | Pretrain | SODA | DailyDialog | MetaICL |
| Pretrain (= RedPajama + LmSys-Chat) | 500k | **-0.55** | -0.22 | -0.74 | -4.9% |
| Pretrain + MetaICL | 500k | -0.59 | -0.26 | -0.82 | -1.2% |
| Pretrain + MetaICL + SODA | 500k | -0.61 | -0.10 | -0.54 | -1.3% |
| Pretrain + MetaICL + SODA | 750k | -0.57 | **-0.09** | **-0.53** | **-1.1%** |

**FLOPS analysis**   Regarding FLOPS, our approach has two notable effects:

- Reduction in attention FLOPS due to the shortened context.
- Computation overhead incurred by the compression process.

The reduction in attention FLOPS becomes more pronounced as the number of processed tokens during inference increases. In Table 12, we compute the minimum length of tokens required to be processed during inference, where the benefits from the shortened context outweigh the compression overhead. Our analysis is based on a context token length of 50, according to the dataset statistics in Table 7. With ⟨COMP⟩ token length of 1, our approach reduces the total computation FLOPS when the length of the processed token during inference surpasses 504. We summarize the results on larger ⟨COMP⟩ token lengths in Table 12.

Table 12: Compression FLOPS overhead analysis on MetaICL with LLaMa-7B. *Threshold* refers to the minimum token length required during inference for the reduction in attention FLOPS to outweigh the compression overhead. We assume that the token length of context $c(t)$ is 50, according to the MetaICL and LaMP datasets' statistics (Table 7).

| | ⟨COMP⟩ token length | | | |
|---|---|---|---|---|
| | 1 | 2 | 4 | 8 |
| Context compression factor | ×50 | ×25 | ×13 | ×6 |
| Threshold (inference token length) | 504 | 1029 | 2148 | 4706 |

**In-depth performance analysis**   We measure the generation performance of our compression approach using the RougeL metric in Table 13. The results verify that our methods deliver the most accurate generation performance compared to other baselines. However, in the case of RougeL, there is a pronounced decrease in performance compared to the full context method, whereas, in the case of accuracy, the performance drop is less than 1%. Upon closer examination of the generated outputs with compressed context, we identify instances where synonyms are generated (e.g., "Different" and "Dissimilar" in the medical_questions_pair task) or variations in letter casing are present (e.g., "Hate" and "hate" in the tweet_eval_hate task). These observations suggest a semantic equivalence between the original and generated results, albeit differences in expression. These findings suggest that our approach performs particularly well in situations where prioritizing preferences or nuances outweighs the need for exact phrasing.

Table 13: Evaluation of RougeL and accuracy metrics with LLaMA-7B on MetaICL test tasks.

| | No context | Full context | Gisting-online | Compressive | CCM-concat | CCM-merge |
|---|---|---|---|---|---|---|
| RougeL | 12.3 | **61.4** | 37.9 | 47.9 | **54.7** | 48.3 |
| Accuracy (%) | 51.7 | **70.8** | 57.7 | 67.8 | **70.0** | 69.6 |

**Design choice of merge function**   In the main experiments, we evaluate an update method based on the arithmetic average of the compressed states up to the present time, *i.e.*, $a_t = 1/t$. Another natural design choice is an exponential moving average (EMA), where $a_t$ is set to a constant

value. This strategy weighs higher importance on recent information compared to the arithmetic average. Table 14 provides a comparison between the arithmetic average and EMA with $a_t = 0.5$, on DailyDialog with LLaMA-7B. The results indicate that both methods yield similar performance. When forming the compression state $h(t)$, our method involves referencing the previous memory Mem$(t\text{-}1)$ (Figure 2). We believe this enables the preservation of overall context, even with exponentially decreasing coefficients for past states by EMA.

Table 14: Comparison of merge function design choices with LLaMA-7B on DailyDialog.

| Method \Time step | 1 | 2 | 4 | 8 | 12 |
|---|---|---|---|---|---|
| EMA | 7.49 | 7.06 | 6.79 | 6.49 | 6.38 |
| Arithmetic average | 7.47 | 7.06 | 6.87 | 6.54 | 6.34 |

**Additional memory-performance graphs** In Figure 10, we present graphs illustrating the relationship between attention KV memory and performance across increasing time steps for MetaICL, LaMP, and DailyDialog. The figure comprehensively compares all methods introduced in our main text, including a fixed-context compression method such as Gisting. From the figure, we verify that our methods exhibit the best memory-performance efficiency. Specifically, our methods achieve superior performance while requiring minimal attention KV memory when compared to existing compression baselines.

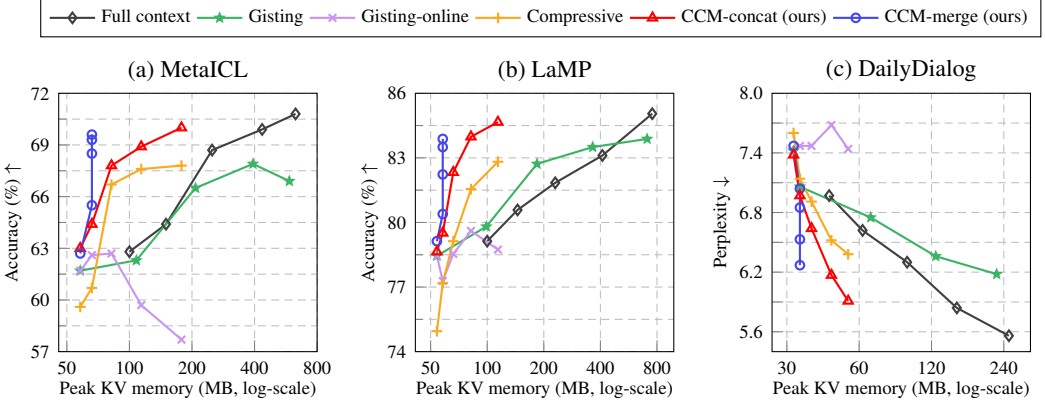

Figure 10: Test performance of methods with LLaMA-7B over increasing time steps in an online inference scenario. The $x$-axis refers to the peak memory space occupied by attention keys/values during compression and inference processes at each time step. Here, the time steps span from 1 to 16, except for DailyDialog, which covers a range of 1 to 12.

### E.2 MODEL AND COMPRESSION TOKEN LENGTH

**Different model architecture** We evaluate our method with an encoder-decoder structured model, Flan-T5-Large (Chung et al., 2022). Since there exists an overlap between the training set of Flan-T5 and the MetaICL dataset (Min et al., 2022), we conduct an evaluation using the LaMP dataset. Table 15 presents the evaluation results at time step 16. While both Gisting and Compressive Transformer exhibit a significant drop in accuracy compared to the full context method, our methods achieve the best performance while requiring less key/value memory on the Flan-T5 architecture.

Table 15: Test accuracy and peak key/value memory size with Flan-T5-Large on LaMP at time step 16. We evaluate performance across five different random seeds for user profile order.

| | No context | Full context | Gisting-online | Compressive | CCM-concat | CCM-merge |
|---|---|---|---|---|---|---|
| Accuracy (%) | $71.1 \pm 0.0$ | $81.8 \pm 0.3$ | $78.4 \pm 0.3$ | $79.7 \pm 0.4$ | $81.9 \pm 0.2$ | **$82.1 \pm 0.3$** |
| Memory (MB) | 20 | 152 | 32 | 32 | 32 | **21** |

**Length of compression token** In Table 16, we analyze the performance of our method across varying compression token lengths. In general, increasing the token length leads to a slight improvement in performance. For MetaICL, we observe a 1% accuracy gain, while the DailyDialog experiment shows a 1% reduction in perplexity as token length increases. However, when comparing our approach to the no-context method, the performance differences attributed to the compression token length are not significant. For example, our method outperforms the no-context approach by approximately 18% in the MetaICL experiment. In our main experiment, we set the compression token length according to the average context length of the target dataset, ensuring consistent compression rates across datasets. We provide detailed configuration values in Table 8.

Table 16: Analysis of ⟨COMP⟩ token length with LLaMA-7B at the maximum time step (Table 7). Here, *concat* refers to CCM-concat, and *merge* denotes CCM-merge.

(a) MetaICL (Accuracy %). No context: 51.6% \ Full context: 70.8%.

|  | ⟨COMP⟩ token length | | | |
|---|---|---|---|---|
|  | 1 | 2 | 4 | 8 |
| concat | 69.5 | 69.3 | 70.0 | **70.0** |
| merge | 68.5 | 68.1 | 68.3 | **69.6** |

(b) LaMP (Accuracy %). No context: 69.5% \ Full context: 85.1%.

|  | ⟨COMP⟩ token length | | |
|---|---|---|---|
|  | 1 | 2 | 4 |
| concat | 84.3 | 83.9 | **84.7** |
| merge | 83.4 | **84.2** | 83.9 |

(c) DailyDialog (Perplexity). No context: 10.3 \ Full context: 5.85.

|  | ⟨COMP⟩ token length | | |
|---|---|---|---|
|  | 1 | 2 | 4 |
| concat | 6.51 | 6.37 | **6.26** |
| merge | 6.67 | **6.62** | 6.63 |

**Larger model scale** In Table 17, we provide evaluation results with LLaMA-13B on MetaICL. Consistent with 7B models, our method exhibits the best performance among the compression baselines while requiring smaller peak attention KV memory.

Table 17: LLaMA-13B test accuracy and peak attention KV memory on MetaICL at time step 16.

|  | No context | Full context | Gisting | Gisting-online | Compressive | CCM-concat | CCM-merge |
|---|---|---|---|---|---|---|---|
| Accuracy (%) | 51.4 | **72.1** | 66.7 | 62.5 | 66.1 | **70.7** | 68.6 |
| Memory (MB) | 78 | 984 | 919 | 278 | 278 | 278 | **103** |

### E.3 COMPARISON TO BASELINES

**Comparison to recurrent compression methods** We conduct a comparative analysis with RMT and AutoCompressor that recurrently compress contexts into token embeddings (Bulatov et al., 2022; Chevalier et al., 2023). These approaches fine-tune OPT pretrained models (Zhang et al., 2022) on the Pile dataset (Gao et al., 2020) to learn compression capabilities. For evaluation, we utilize the fine-tuned models available on the official GitHub repository[1]. We conduct separate experiments on each baseline because the released RMT and AutoCompressor models show different performances without compression (AutoCompressor in Table 18 and RMT in Appendix Table 22). For a fair comparison, we also provide fine-tuned results of the baseline models on MetaICL training tasks using identical training steps to ours, denoted as *AutoCompressor-finetune* and *RMT-finetune*. As shown in the tables, our compression methods demonstrate superior performance and efficiency. Specifically, RMT and AutoCompressor necessitate recursive model computation at each training step, incurring significant computation time. As shown in Table 18, AutoCompressor requires approximately **7× longer training time** per sample than our approach. Meanwhile, our methods exhibit superior performance while using less key/value memory, demonstrating its effectiveness.

Table 18: Comparison with AutoCompressor OPT-2.7B on MetaICL test tasks at time step 16. We measure the training time using identical samples on an A100 GPU. We evaluate performance across five different random seeds for demonstration order.

|  | No context | Full context | AutoComp. | AutoComp.-finetune | CCM-concat | CCM-merge |
|---|---|---|---|---|---|---|
| Accuracy (%) | 41.4 ± 0.0 | **54.2** ± 0.5 | 48.1 ± 0.5 | 50.9 ± 0.4 | **53.5** ± 0.5 | 52.3 ± 0.3 |
| Peak KV memory (MB) | 31 | 394 | 156 | 156 | 111 | **41** |
| Training time per sample (ms) | - | - | 1330 | 1330 | **195** | **195** |

---

[1] https://github.com/princeton-nlp/AutoCompressors

**Comparison to fixed-context compression**    In Table 19, we present evaluation results of Gisting with the fixed-context compression setting described in Figure 5-b. While having the same inference complexity as CCM-merge, the fixed-context setting incurs significant memory demands during compression. On the other hand, our approach maintains minimal memory requirements for both stages, having a low peak memory usage. Moreover, our method improves the performance by 3%p compared to Gisting, validating the effectiveness of our training strategy in online scenarios.

Table 19: Comparison to a fixed-context compression method (Gisting) with LLaMA-7B on MetaICL test tasks at time step 16.

|  | Full context | Gisting | CCM-concat | CCM-merge |
|---|---|---|---|---|
| Accuracy (%) | **70.8** $\pm$ 0.1 | 66.9 $\pm$ 0.2 | **70.0** $\pm$ 0.2 | 69.6 $\pm$ 0.1 |
| Peak KV memory (MB) | 630 | 588 | 178 | **66** |

**Comparison to text summarization**    MemoryBank proposes reducing context size through text summarization during language model interaction (Zhong et al., 2023). However, this approach comes with additional computational costs for summarization and the overhead of processing the summarized text for subsequent inference. In contrast, our approach allows for more efficient inference without the aforementioned overhead by caching key/value pairs of compression tokens. Following MemoryBank, we conduct experimental comparisons with LLaMA-7B on DailyDialog. Specifically, we use the summarization prompt from MemoryBank to compress context through OpenAI gpt-3.5-turbo API (ChatGPT) and then evaluate models with summarized contexts. Table 20 shows the test perplexity of methods. The results confirms that our approach achieves superior performance with smaller context memory size, demonstrating the effectivness of our key/value compression approach.

Table 20: Comparison to a text summarization method with LLaMA-7B on the DailyDialog test set.

|  | No context | Full context | MemoryBank | CCM-concat | CCM-merge |
|---|---|---|---|---|---|
| Perplexity | 10.6 | 5.59 | 7.06 | **5.98** | 6.34 |
| Compressed context length | 0 | 222 | 60 | 24 | **2** |

Table 21: Evaluation results of default LoRA and our conditional LoRA with LLaMA-7B.

(a) LaMP (Accuracy %)

|  | Default | Conditional (ours) |
|---|---|---|
| CCM-concat | 83.9 | **84.7** |
| CCM-merge | 82.6 | **83.9** |

(b) DailyDialog (Perplexity)

|  | Default | Conditional (ours) |
|---|---|---|
| CCM-concat | 6.01 | **5.96** |
| CCM-merge | 6.42 | **6.33** |

Table 22: Comparison with RMT OPT-2.7B on MetaICL at time step 16. We measure the training time using identical samples on an A100 GPU. We evaluate performance across five different random seeds for demonstration order.

|  | No context | Full context | RMT | RMT-finetune | CCM-concat | CCM-merge |
|---|---|---|---|---|---|---|
| Accuracy (%) | $42.1 \pm 0.0$ | $54.5 \pm 0.4$ | $44.4 \pm 0.4$ | $50.0 \pm 0.3$ | $52.3 \pm 0.4$ | $52.2 \pm 0.3$ |
| Peak KV memory (MB) | 31 | 394 | 63 | 63 | 111 | **41** |
| Training time per sample (ms) | - | - | 1330 | 1330 | **195** | **195** |

Table 23: Test accuracy (%) on MetaICL with LLaMA-7B. The test set is identical across time steps.

| Time step | No context | Full context | Gisting-online | Compressive | CCM-concat | CCM-merge |
|---|---|---|---|---|---|---|
| 1 | 51.7 | 62.8 | 61.7 | 59.6 | 63.0 | 62.7 |
| 2 | 51.7 | 64.4 | 62.6 | 60.7 | 64.4 | 65.5 |
| 4 | 51.7 | 68.7 | 62.7 | 66.7 | 67.8 | 68.5 |
| 8 | 51.7 | 69.9 | 59.7 | 67.6 | 68.9 | 69.3 |
| 16 | 51.7 | 70.8 | 57.7 | 67.8 | 70.0 | 69.6 |

Table 24: Test accuracy (%) on LaMP with LLaMA-7B. The test set is identical across time steps.

| Time step | No context | Full context | Gisting-online | Compressive | CCM-concat | CCM-merge |
|---|---|---|---|---|---|---|
| 1 | 69.5 | 79.1 | 78.5 | 75.0 | 78.6 | 79.1 |
| 2 | 69.5 | 80.6 | 77.3 | 77.2 | 79.5 | 80.4 |
| 4 | 69.5 | 81.8 | 78.5 | 79.1 | 82.3 | 82.2 |
| 8 | 69.5 | 83.1 | 79.6 | 81.6 | 84.0 | 83.5 |
| 16 | 69.5 | 85.1 | 78.7 | 82.8 | 84.7 | 83.9 |

Table 25: Test perplexity on DailyDialog with LLaMA-7B.

| Time step | No context | Full context | Gisting-online | Compressive | CCM-concat | CCM-merge |
|---|---|---|---|---|---|---|
| 1 | 8.93 | 6.97 | 7.42 | 7.60 | 7.38 | 7.47 |
| 2 | 9.06 | 6.62 | 7.47 | 7.14 | 6.97 | 7.04 |
| 4 | 9.33 | 6.30 | 7.47 | 6.91 | 6.64 | 6.85 |
| 8 | 9.85 | 5.84 | 7.68 | 6.52 | 6.17 | 6.53 |
| 12 | 9.67 | 5.56 | 7.44 | 6.38 | 5.91 | 6.27 |

