# OpenReview forum: "Compressed Context Memory for Online Language Model Interaction"
_ICLR.cc/2024/Conference — ICLR 2024 poster_

### Official Review · Reviewer_Svka · 2023-10-28

**Soundness:** 3 good
**Presentation:** 3 good
**Contribution:** 3 good
**Rating:** 6
**Confidence:** 4

**Summary:**

This paper proposes a way to compress LLM context in an online setting, such as conversation, where more information could arrive and needs to be compressed. Here are the contributions:
1. Extended the gisting technique to multi-context settings, and demonstrated the gains are better than naively adapting gisting to this setting. Further, the authors show that the proposed technique has very little gap with no compression, on three datasets.
2. The authors also have proposed a parameter efficient method for training this mechanism, which doubles as a regularization. They also proposed an efficient way to train the model, by parallelizing the originally sequential context segments.
3. Finally there are ablation studies on various design choices, such as the conditional adapter, the model architecture etc.

**Strengths:**

The exact method proposed is novel. On the three presented datasets, it seems to work pretty well, judging from the small gap between compressed and uncompressed setups. The problem this paper is trying to address is very important to the field, and I believe this work has significant contributions. Finally, the paper is generally easy to read.

**Weaknesses:**

This method could be evaluated on more diverse datasets, such as those used for long context, by utilizing a sliding window for example. From my perspective, this work has the potential to be applied more broadly beyond ICL, or dialog. It'll also be nice to have the comparison with RMT and AutoCompressor in the main text, as they are very relevant for this problem.

**Questions:**

1. In eq (1), $h(t)$ is conditioned on $Mem(t - 1)$. However, in figure 4, and the description of the parallelized training procedure, it's unclear how this could be done, as all the $Mem(\*)$ are computed after $h(\*)$, for the same layer. Do you condition $h(t)$ on the previous layer's $Mem(t - 1)$? If so please update the text to make it obvious.
2. The motivation for the conditional adapter is to regularize against overfitting. The authors only reported the training loss of LLAMA 7B without context. To make it more clear on the overfitting situation, could you also report the loss with context, and the test performance with and without context.
3. The masking doesn't seem to be autoregressive in figure 4b?
4. Table 3 only has numbers on MetaICL. What about the other two datasets?
5. In section 4.2, on different model architecture, have you tried non instruction tuned T5? From what I can tell, instruction following is not necessary for the three datasets used in the paper.

---

> ### Author Response · Authors · 2023-11-19
> **Author Response**
>
> We thank you for your valuable efforts and time in providing insightful feedback on our work. We would like to address questions below.
>
> **Q. Evaluation on more diverse datasets**
> - Thank you for your question. We would like to first mention that MetaICL consists of 26 test datasets, covering a variety of tasks such as reading comprehension, question answering, and natural language inference. As you mentioned, exploring the applicability of our method beyond ICL and dialog to more general scenarios is indeed practically significant. To investigate the potential for generalization in our approach, we train a compression module using MetaICL train tasks and SODA, a conversation dataset. Then we evaluate the model across various scenarios, including MetaICL test tasks, LaMP, and DailyDialog.
> - For detailed experiment results, please refer to the **general response above**. The results demonstrate that our compression method possesses the ability to generalize to new datasets or scenarios. We will incorporate this analysis into our paper.
>
> **Q. Have the comparison with RMT and AutoCompressor in the main text**
> - Thank you for your comments. We will make sure to incorporate the comparison analysis from the Appendix into the main text.
>
> **Q. Eq 1 and Figure 4. All the Mem($\cdot$) are computed after h($\cdot$), for the same layer. Do you condition h(t) on the previous layer's Mem(t-1)?**
> - We apologize for any confusion caused. Equation 1 signifies that $h(t)$ (keys/values at all layer) is obtained through attention on the keys/values of $\text{Mem}(t-1)$. Specifically, the key/value at each layer is obtained after the attention process from the previous layer. Similarly, in the parallelized operation in Figure 4, the key/value at each layer of $h(t)$ is obtained through the key/value of $\text{Mem}(t-1)$ from the previous layer. We will add this clarification to the main text.
>
> **Q. On overfitting situation, could you also report the loss with context, and the test performance with and without context?**
> - We would like to share the results regarding the points you raised. As mentioned in our paper, training the model using the standard LoRA for compression led to overfitting, evident from the decrease in the training loss without context from an initial 2.69 to 1.84 after training. The **training loss with compressed context** for this model is 0.58, indicating that the model has indeed learned compression capabilities on the training set.
> - However, the **test loss with compressed context** is 1.01 indicating the existence of generalization gap for compression. On the other hand, the **test loss without context** is 2.89. Considering the training loss without context of 1.84, the large gap between the train/test losses suggests the presence of overfitting to the input I(t). We will include this analysis in the paper.
>
> **Q. The masking doesn't seem to be autoregressive in figure 4b?**
> - Thank you for bringing this to our attention. It appears that the confusion arose from separating memory and context parts in the visualization of the attention mask. The calculation of memory $\text{Mem}(t-1)$ occurs after $c(t-1)$ and its subsequent <COMP> token. Reordering the top row of the figure 4b to align with this temporal relation will yield an autoregressive mask. We will adjust the figure to enhance clarity.
>
> **Q. Table 3 with other two datasets**
> - We conduct an ablation study for our conditional LoRA on a different dataset, and the results are as follows:
> | Dataset | Method | Default LoRA | Conditional LoRA (ours) |
> |---|---|---:|---:|
> | DailyDialog (Perplexity) | CCM-concat | 6.01 | 5.96 |
> | DailyDialog (Perplexity) | CCM-merge | 6.42 | 6.33 |
> | LaMP (Accuracy %) | CCM-concat | 83.9 | 84.7 |
> | LaMP (Accuracy %) | CCM-merge | 82.6 | 83.9 |
>
> - From the above results, we observe that our conditional LoRA is effective across various datasets. We will include this results in the revision.
>
> **Q. Results on T5 without instruction finetuning**
> - Thank you for your advice. As you suggested, we conduct LaMP experiments on T5-Large without instruction tuning, and the results are as follows:
> |  | No-context | Full-context | Gisting-online | Compressive | CCM-concat | CCM-merge |
> |---|---|---|---|---|---|---|
> | Accuracy (%) | 65.9 | 81.7 | 77.6 | 79.0 | 81.7 | 80.6 |
>
> - From these results, we observe that our compression method remains effective in this model. Notably, the performance improvement by the utilization of compressed context, compared to no-context, is 16%, surpassing the 10% improvement observed in the case of Flan-T5 (Table 5). We will incorporate this result into the revised version.
>
> Thank you once again for the valuable feedback. If you have any remained questions, please let us know.

---

> > ### Author Response · Authors · 2023-11-23
> > **Discussion Closure and Request for Confirmation**
> >
> > Dear reviewer, we want to remind you that the discussion period is coming to a close soon. If you have any further questions or if there are additional points you would like to discuss, please feel free to let us know.
> >
> > Thank you once again for your time and effort. We appreciate your consideration.
> >
> > Best regards,
> > Authors

---

### Official Review · Reviewer_UgoG · 2023-10-30

**Soundness:** 2 fair
**Presentation:** 3 good
**Contribution:** 2 fair
**Rating:** 5
**Confidence:** 4

**Summary:**

This paper proposes an interesting method to compress context memory. Compared with previous context compression work, the main novelty and contribution claimed in this paper is its ability to handle dynamic contexts. Its idea is similar to RMT, letting the transformer to work in a recurrent memory mechanism. By inserting a compression token in the in-context manner like Gist, Autocompressor and ICAE, the method can process the context recursively into a constant memory cost. The experiments conducted on Meta-ICL, LaMP and DailyDialog dataset show a promising compression rate with full-context performance.

**Strengths:**

- A interesting method to compress contexts in the few-shot learning setting.

- The results evaluated in the few-shot learning tasks show the effectiveness and superiority over the conventional approaches like RMT and Gist.

**Weaknesses:**

While this paper presents a seemingly promising solution to long contexts, I have significant concerns about several limitations.

Firstly, one of the main focuses of this paper is handling dynamic context for interaction. Judging from its experimental design, it mainly conducts experiments with a fine-tuned LLM for few-shot learning scenarios, which are generally simpler tasks, all being multi-choice, or classification tasks. The methods primarily compared in this paper are general context compression or long context handling methods (general and generative tasks). This involves 3 issues. The first issue: for simpler tasks like classification, compression is relatively easy (this is why previous models were easily distilled but GPT was not; I believe it's not because GPT is hard to distill, but because GPT is not for a specific task, but a general model. The authors may better know what I said if reading the paper about information bottleneck: https://arxiv.org/pdf/1503.02406.pdf). Therefore, it's not surprising that this method achieves good compression results (model size compression and context compression are similar, both reducing model capacity), but I believe it's hard for this paper's method and results to scale to general scenarios. At least in this paper, I didn't see any general tests to prove its compression effect. The second issue: for the few-shot learning setting, adding compression tokens after the demonstration makes sense, but for general scenarios, there is no definite boundary to limit compression tokens, making this method hard to generalize. Even if a compression token can be added every K tokens, this approach would lead to inefficient training, as a large amount of sampling is required to ensure the model learns well for each position. The third issue: the main setting of this paper is few-shot learning, and its main claim is online interaction. But for few-shot learning, there doesn't seem to be any online interaction. Users usually provide all demonstrations at once for the model to give an answer, and it's hard for me to imagine a setting where users incrementally provide demonstrations to the model.

Secondly, for dialogue tasks, this is a context compression for a specific task (more preciesely, for a specific dataset). Similar to what I mentioned above, if it's for a specific task/benchmark, there is actually a lot of compression space, which has been discussed in many previous works, such as: https://arxiv.org/pdf/2301.12726.pdf. For context compression of a specific task, even without this method, other methods should also achieve good compression results.

Thirdly, I am not entirely convinced by the results presented in the paper. For example, in Table 15, the performance of RMT is almost the same as that of No context, which is hard to believe. Moreover, Table 15 is an experiment conducted on OPT-2.7b. The few-shot learning ability of the 2.7b OPT model, as far as I understand, should be very weak, and changes in the order of sample arrangement will significantly affect the results. For a method like this paper's, which is similar to a recurrent method, it should be very easily influenced by later samples. Unfortunately, I didn't see any discussion about this.

Some questions:
1. Figure 4 is a little confusing. In (a), what do the blocks in different colors mean?
2. It seems that this work uses 1 single 80GB A100 for training. Could the authors provide more details about the fine-tuning process?

**Questions:**

See the weakness section

---

> ### Author Response · Authors · 2023-11-19
> **Author response (1)**
>
> We thank you for your valuable efforts and time in providing insightful feedback on our work. We would like to address questions below.
>
> **Q. General tests to prove its compression effect**
> - Thank you for the pointer. To demonstrate the generalization ability of our method in more general scenarios, we train a single compression model and evaluate its performance across various datasets. Specifically, we leverage **MetaICL training tasks and the conversation dataset, SODA, as our training data**, and then conduct evaluation on **multiple test scenarios: MetaICL test tasks, LaMP, and DailyDialog**.
> - For detailed experiment results, please refer to the **general response above**. The results demonstrate that our compression method possesses the ability to generalize to new datasets or scenarios. We will incorporate this analysis into our paper.
>
> **Q. For few-shot learning, there doesn't seem to be any online interaction**
> - Thank you for your insights. We argue that in-context learning scenarios exist where not all demonstrations are provided initially. For instance, in personalization scenarios, examples of user patterns or preferences can be collected through online interactions.
> - Moreover, in cases involving a large number of tasks, as discussed in recent work such as "*Large-scale Lifelong Learning of In-context Instructions and How to Tackle It*, ACL 2023," it is conceivable to consider scenarios where users continuously add demonstrations as data is collected.
> - Alternatively, inspired by "*Active Example Selection for In-Context Learning*", EMNLP 2022, we can imagine a system where users set the goal for a specific task, and the language model learns through interaction by querying the user or internet, allowing for continuous improvement without the user having to prepare multiple demonstrations. We believe that MetaICL and LaMP, evaluating the in-context learning performance for new tasks/users, are effective evaluation datasets for the aforementioned scenarios.
>
> **Q. For a specific task/benchmark, there is actually a lot of compression space**
> - Thank you for your comments. As you pointed out, we believe that for specific tasks, the compression space is larger, making compression more manageable. This means that task-specific compression methods can achieve higher compression efficiency compared to general compression methods. In some practical industrial settings, there are attempts to fine-tune foundation models on individual datasets to obtain specialized models (https://platform.openai.com/docs/guides/fine-tuning). In a similar vein, task-specific compression could be applied to achieve the better compression capability.
> - It is noteworthy that **our method is task-agnostic**. We can apply our approach to a wide range of scenarios, including various forms of data such as multi-tasks in MetaICL, personalization, and dialogues, without requiring any specific modifications. We believe obtaining a task-specific compression module solely based on data without task-specific knowledge or modification is practically meaningful.
> - Lastly, we verify that our approach does not overfit to specific datasets or benchmarks as shown in the generality test mentioned above. Notably, the compression model trained on the SODA dataset maintains effective compression performance on the DailyDialog dataset, demonstrating its versatility. We will include these discussions in the paper for further clarity.
>
> **Q. A large amount of sampling is required to ensure the model learns well for each position**
> - Thank you for your feedback. Our compression method **operates recurrently using the same embedding and model weights for each compression step**, without utilizing different parameters conditioned on time steps or positions. Specifically, compression tokens at different time steps share the same embedding. The gradients computed for the loss during the training process are simultaneously backpropagated to all tokens across all time steps.
> - Our approach effectively learns without the need for individual sampling for each position (time step). To illustrate, we conduct a comparison between training CCM-concat by sampling time steps across [1,...,16] and training with a fixed time step of T=16 in MetaICL with LLaMA-7B. The table below shows the test accuracy of two models across a range of time steps.
> | Time step | 1 | 4 | 16 |
> |-|-:|-:|-:|
> | Time sampling | 63.0 | 67.8 | 70.0 |
> | Only T=16 | 62.5 | 67.4 | 70.1 |
>
> - The results indicate that there is not a significant difference between the two methods. This suggests that substantial sampling for compression at each time step or position is not necessary for our approach. We will include this analysis in the revision.

---

> ### Author Response · Authors · 2023-11-19
> **Author response (2)**
>
> **Q. Regarding performance of RMT in Table 15**
> - Thank you for your question. The experiments in Table 15 were conducted following the procedure outlined below. Firstly, we obtained the OPT full attention-2.7b-4k model, publicly available from the AutoCompressor GitHub. We fine-tuned this model on the MetaICL training set for ICL meta-training, following the settings in Table 9. Subsequently, we performed compression fine-tuning for RMT, AutoCompressor, and CCM methods, with the same number of steps. The training of RMT and AutoCompressor was carried out using the official AutoCompressor GitHub code. This ensured that all experiments were conducted **using the same pretrained model and the same training data**. We note that the publicly available RMT and AutoCompressor compression models on GitHub show **different ICL performances without compression**.
> - We acknowledge the approach mentioned above does not assess the original compression capabilities of RMT and AutoCompressor, both trained on a vast corpus comprising billions of tokens. We speculate that the lower performance measured for RMT in Table 15 is due to insufficient training. To address this issue, we obtain the RMT and AutoCompressor compression models from the AutoCompressor GitHub and conduct evaluations on these baseline models. Since these two models have different ICL performances, we conduct separate experiments. For a fair comparison, we also provide finetuned results of the baseline models using the identical training steps to ours. It’s noteworthy that our method only optimizes conditional LoRA for compression and **does not affect the ICL performance of the baseline models**.
>
> ||No context|Full context|RMT|RMT+ finetune|CCM-concat|CCM-merge|
> |-|-:|-:|-:|-:|-:|-:|
> |Accuracy (%)|42.1|54.5|44.4|50.0|52.3|52.2|
> |Peak KV memory (MB)|31|394|63|63|111|41|
> |Training time per batch (s)|-|-|-|170|25|25|
>
> ||No context|Full context|AutoComp.|AutoComp.+ finetune|CCM-concat|CCM-merge|
> |-|-:|-:|-:|-:|-:|-:|
> |Accuracy (%)|41.4|54.2|48.1|50.9|53.5|52.3|
> |Peak KV memory (MB)|31|394|156|156|111|41|
> |Training time per batch (s)|-|-|-|170|25|25|
>
> - The first table shows the results by using the RMT model from the GitHub and the table below shows the results by using the AutoCompressor as the full-context model. The results above show a decrease of approximately 5% in both No-context and Full-context performance compared to Table 15, due to the lack of ICL meta-training of baseline models. However, even in this setting, our compression method demonstrates superior efficiency. We acknowledge that the explanation for Table 15 in our paper was insufficient and, to ensure more accurate comparison, we will incorporate the above description and results into the revised version.
>
> **Q. Changes in the order of sample arrangement**
> - Thank you for the pointer. To measure sensitivity on the order of demonstrations, we fix the demonstration set of each test task and evaluate performance by changing only the demonstration order using random seeds. The table below shows the test accuracy on MetaICL with OPT-2.7B over various random seed for the demonstration order.
> |Model \ Seed|0|1|2|3|4|Stdev.|
> |-|-|-|-|-|-|-:|
> | Full-context|59.1|59.8|59.6|59.3|60.5|0.54|
> | CCM-concat | 53.1 | 52.0 | 52.9 | 52.0 | 52.7 | 0.55 |
> | CCM-merge | 53.2 | 53.2 | 53.5 | 53.1 | 53.3 | 0.15 |
> - In the case of LLaMA-7B, we obtain Standard deviations under 0.25 for all three methods. As evident from the results above, the OPT model shows a larger performance variance based on the order compared to LLaMA-7B. However, it is noteworthy that the variance based on order is not significantly larger compared to the performance difference between methods. Specifically, CCM-Concat outperforms other compression baselines by over 2% in test accuracy (Table 15). We will incorporate these findings into the revision.
>
> **Q. In Figure 4 (a), what do the blocks in different colors mean?**
> - In the figure, each block represents a key/value pair. Blocks in yellow correspond to keys/values of memory, those in red represent keys/values of compression tokens, and those in green correspond to key/value pairs associated with regular text. We will add this explanation in the revision.
>
> **Q. More details about the fine-tuning process**
> - We provide the fine-tuning process in Appendix C. During the fine-tuning process, only the LoRA adapter is additionally trained, following the training settings outlined in Table 9, which is adopted from the Gisting paper. A batch size of 128 are processed on a single GPU through gradient accumulation, with training ranging from 300 to 2000 steps depending on the dataset size. The training time takes around 3 to 24 hours. For a fair comparison with compression baselines considered in the paper, we conducted the comparison after fine-tuning all methods with the same process.
>
> Thank you once again for the valuable feedback. If you have any remained questions, please let us know.

---

> > ### Author Response · Authors · 2023-11-23
> > **Discussion Closure and Request for Confirmation**
> >
> > Dear reviewer, we want to remind you that the discussion period is coming to a close soon. If you have any further questions or if there are additional points you would like to discuss, please feel free to let us know.
> >
> > Thank you once again for your time and effort. We appreciate your consideration.
> >
> > Best regards,
> > Authors

---

### Official Review · Reviewer_JvdE · 2023-10-30

**Soundness:** 3 good
**Presentation:** 3 good
**Contribution:** 3 good
**Rating:** 6
**Confidence:** 4

**Summary:**

In the LLMs era, during long-term online user-model interactions, a lot of past dialogue history cannot fit the context length of the model and thus have to be compressed into memory to ensure that the model can memorize the past dialogue. It is an emerging and interesting research question. This paper introduces a novel method to compress the long context memory in the online user-machine interactions, named CCM as well as two variants, merge and concat. As LLM is hard to fine-tune to engage such CCM, the author also adopts LoRA adapter to fine-tune LLM in a lightweight setting to engage the model to learn to compress the memory.

**Strengths:**

1. The paper is overall sound. The method design is concise, effective, and efficient. Compared with retrieval-based method to re-compute the sentence embedding, the CCM can directly adopt the KV cache of introduced <COMP> token as the memory vector for one utterance and utilize them in further inference. To engage the LLM to utilize such CCM, the parallel training and LoRA adapter are designed well for efficient adaptation.

2. The CCM is efficient in both training and inference. Firstly, there is no need to re-compute the sentence embedding and instead caching the attention keys and values. Secondly, the memory storage cost, the compression ratio, and the algorithmic complexity all demonstrate the efficiency of the method. Thirdly, LoRA based adapter tuning brings a lot of efficiency in memory-engaged adaptation.

2. The evaluation is comprehensive and diverse. Three important benchmarks, MetaICL, LaMP, and DailyDialog are selected for evaluation and MetaICL covers 26 tasks with high-diversity.

**Weaknesses:**

1. The CCM method is not that novel and has been explored well in some important early milestones before the creation of Transformer, i.e., Memory Networks, Fast Weights to Attend Recent Past. The author should mention and discuss the relation with these methods. Additionally, the Compress Transformer should be briefly introduced as it is not a universally known preliminary for readers.

2. In terms of the baselines, in the main tables, CCM is only compared with “no context" and "full context" baselines on accuracy, which lacks sufficient comparison for demonstrating the effectiveness of the method. As least, the retrieval-based method should be considered as an important baseline and it is now the universally-adopted method for memory compression. If I understand the paper correctly, the token embedding produced by <COMP> token is the same as a sentence embedding. The author can follow MemoryBank for adopting retrieval-based method to compress the long-context during online interactions. Other baselines like LongMem and UnlimitedFormer can be also considered but not necessary.

**Questions:**

1. For Table 4 and 5, can you add a new row to present the memory size using number of tokens, which is a more intuitive metric than the disk size in MBs?

2. For CCM-merge, it is a vanilla average of all memory features, which does not make sense. Typically, the latest memory might be more important and contribute more to the current dialogue turn. Is it possible to add a weighted coefficient in terms of time steps for a weighted merge on all memory features, i.e. $$Softmax([1:t_i])\in R^t$$?

3. Missing references which have been mentioned in weakness section.

**Details Of Ethics Concerns:**

None.

---

> ### Author Response · Authors · 2023-11-19
> **Author response**
>
> We thank you for your valuable efforts and time in providing insightful feedback on our work. We would like to address questions below.
>
> **Q. Missing references**
> - Thank you for your mention. The papers you mentioned, "Memory Networks" and "Fast Weights to Attend Recent Past," are indeed important seminal works, and we will make sure to add a comparison to related work in our paper. It's worth noting that our approach differs in that we propose a compression adapter to the existing foundation model, whereas they propose entirely new models. As you pointed out, we will also include an explanation of the Compressive Transformer in the preliminary section to enhance the reader's understanding.
>
> **Q. Table with number of tokens**
> - Thank you for the suggestion. As you mentioned, we will add the corresponding row. The results for MetaICL are as follows (Table 4).
> | Method | No-context | Full-context | Gisting | Gisting-online | Compressive | CCM-concat | CCM-merge |
> |-|-:|-:|-:|-:|-:|-:|-:|
> | Max # KV token during inference | 50 | 630 | 588 | 178 | 178 |  178 |  66 |
>
> **Q. More sophisticated memory update mechanisms**
> - It’s a great idea! Based on your advice, we have formulated a more **general form of the memory update** equation as follows:
> $\text{Mem}(t) = (1 - a_{t-1}) \text{Mem}(t-1) + a_{t-1} h(t), \text{for}\ a_{t-1} \in [0,1].$
> - In our paper, we proposed an update method based on the arithmetic average (AA) of the compressed state up to the present time, i.e., $a_{t-1}=1/t$. Another natural design choice can be an exponential moving average (EMA), where $a_{t-1}$ is set to a constant value. This strategy weigh higher importance on recent information, when compared to the arithmetic average. In below, we train a model with EMA and compare it's test performance to the original CCM-merge (AA).
> | Time step | 1 | 2 | 4 | 8 | 12 |
> |-|-:|-:|-:|-:|-:|
> | EMA | 7.49 | 7.06 | 6.79 | 6.49 | 6.38 |
> | AA | 7.47 | 7.06 | 6.87 | 6.54 | 6.34 |
>
> - The table above provides perplexity results of the arithmetic average and EMA with $a_{t}=0.5$, in DailyDialog with LLaMA-7B. S The results indicate that both methods yield similar performance. When forming the compression state $h(t)$, our method involves referencing the previous memory, $\text{Mem}(t-1)$. We believe this enable preservation of overall context, even with exponentially decreasing coefficient for past states by EMA. We also note that CCM-concat can be interpreted as dynamically inferring coefficients for individual hidden states $h(t)$ through the transformer attention mechanism.
>
> **Q. Comparison to retrieval-based approach**
> - Thank you for your feedback. As you pointed out, memory retrieval-based approaches are indeed important baselines for our work. Existing studies like Memorizing Transformer, and recent studies such as MemoryBank, LongMem, and UnlimitedFormer primarily focus on the retrieval process and training of retrieval-based models, with less emphasis on memory compression. These retrieval studies adopt strategies to cache key-value pairs for individual tokens and assume the existence of extensive storage. However, as seen in Table 4 and 5 of our paper, LLM's keys and values demand a significant amount of storage, even for context size of 1024 tokens, reaching several hundred megabytes. In scenarios like user-level personalization and conversation systems, such high storage requirements can become problematic. We will add these references in the related work section.
> - We also note that MemoryBank proposes reducing context history size through summarization. However, this approach comes with **additional computational costs for summarization and the overhead of processing the summarized text** for subsequent inference. In contrast, our approach allows for more efficient inference without the aforementioned overhead, thanks to the caching of key-value pairs of compression tokens. Following the MemoryBank paper, we conduct experimental comparisons with LLaMA-7B on DailyDialog. Specifically, we use the **summarization prompt from the paper** to compress context through OpenAI gpt-3.5-turbo API (ChatGPT) and then evaluate models with summarized contexts. The table below compares the test perplexity of methods.
> |  | No-context | Full-context | MemoryBank (summarization) | CCM-concat | CCM-merge |
> |-|-:|-:|-:|-:|-:|
> | Perplexity | 10.6 | 5.59 | 7.06 | 5.98 | 6.34 |
> | Compressed context length | 0 | 222 | 60 | 24 | 2 |
>
> - From the above results, we can confirm that our approach achieves superior performance with less compression overhead and shorter compressed context length. In contrast to text summarization methods that compress context into text form, our approach achieves better compression efficiency by compressing information into key/value parameter space. We will incorporate these findings into the paper.
>
> Thank you once again for the valuable feedback. If you have any remained questions, please let us know.

---

> > ### Comment · Reviewer_JvdE · 2023-11-21
> > **Response to Rebuttal**
> >
> > Thanks for the great response!
> >
> > Firstly, it is glad to see that a better average mechanism brings a performance improvement to your method.
> >
> > Secondly, thanks for the clarifications and improved presentation to help the readers get better understanding towards your paper.
> >
> > Thirdly, the comparisons with the retrieval-based methods resolve my concern and help consolidate the effectiveness of the method.
> >
> > I will keep my rating to 6 of acceptance.

---

> > > ### Author Response · Authors · 2023-11-21
> > >
> > > Thank you for the encouraging comments!
> > >
> > > Best regards,
> > > Authors

---

### Official Review · Reviewer_JGpv · 2023-10-31

**Soundness:** 3 good
**Presentation:** 3 good
**Contribution:** 3 good
**Rating:** 6
**Confidence:** 4

**Summary:**

The paper proposes a novel compressed context memory system for dynamically compressing contextual information during online inference with language models. This allows the model to handle continually expanding contexts efficiently.

The main contributions are:

- A compressed memory framework that condenses context into a compact representation which is dynamically updated during online inference. This reduces memory usage and computation compared to using the full context.

- A parallel training strategy using masked attention to learn the context compression in a single forward pass.

- A conditional adapter applied only to compression tokens to avoid overfitting to inputs during training.

**Strengths:**

- The problem of efficiently handling expanding contexts is highly relevant given the online nature of systems like ChatGPT. The paper addresses an important open challenge.

- The method is flexible and broadly applicable to diverse online inference scenarios like multi-task learning, personalization and conversation.

- Empirical evaluations across three datasets substantiate the memory and computation advantages over baselines. The method achieves slightly lower performance than the full context with 5x smaller memory.

- The parallel training strategy is effective in enabling large model optimization. The conditional adapter improves compression capability.

- The complexity analysis clearly articulates the efficiency benefits, and ablation studies validate the design choices.

**Weaknesses:**

- The main limitation of the proposed compression framework is that it is task-specific. The compression module must be trained for each task, which requires additional data, computation, and cannot generalize to new tasks. This is a significant drawback in the context of foundation models which are trained on large datasets for general-purpose use.
- There is still a obvious gap in performance between the compressed and full context models. The paper does not provide a clear explanation for this gap. The authors should provide more analysis into why the compressed context is less effective.
- While the compression framework is novel, the proposed memory update functions are basic. More sophisticated memory update mechanisms could further enhance capability.
- The comparison is primarily with simple adaptations of fixed-context compression methods. A direct comparison to recurrent memory approaches, such as linear Transformers, would be more informative.

**Questions:**

1. **Task-Specific Limitation**: The framework is mentioned to be task-specific and requires retraining for each new task. Could you elaborate on how this limitation affects the scalability of the proposed method, especially in real-world applications where diverse tasks are common? Additionally, are there any plans or potential strategies to make the framework more task-agnostic?

2. **Additional Resources for Training**: Given that the compression module necessitates extra data and computational resources for training on each task, can you provide a quantitative analysis of the additional resources required compared to other existing methods? How does this additional overhead impact the practicality of adopting your framework, particularly in resource-constrained environments?

3. **Inability to Generalize**: The framework's inability to generalize to new tasks could be a significant disadvantage. Have the authors considered hybrid approaches that combine task-specific and task-agnostic components to mitigate this limitation? If so, what were the challenges or outcomes of these considerations?

4. **Performance Gap Analysis**: The paper notes a clear performance gap between the compressed and full context models. Could the authors provide a more in-depth analysis or hypotheses as to why this gap exists? Are there particular types of data or tasks where this performance gap is more pronounced?

5. **Basic Memory Update Functions**: The memory update functions in the proposed framework are described as basic. Could the authors provide examples or discussions on more sophisticated memory update mechanisms that could potentially enhance the framework’s capabilities? What prevented the integration of these more advanced mechanisms in the current version of the framework?

6. **Lack of Comparison to Recurrent Memory Approaches**: The comparison in the paper is mainly drawn with simple adaptations of fixed-context compression methods. Could the authors justify the choice of these particular baselines and discuss the reasons for not including a direct comparison with recurrent memory approaches, such as linear Transformers? How might the results differ with these additional comparisons?

7. **Explaining Effectiveness of Compressed Context**: In relation to the performance gap, could the authors shed light on any specific scenarios or data types where the compressed context performs particularly well or poorly? Understanding these nuances could help in better positioning the framework and identifying areas for improvement.

---

> ### Author Response · Authors · 2023-11-19
> **Author response (1)**
>
> We thank you for your valuable efforts and time in providing insightful feedback on our work. We would like to address questions below.
>
> **Q: Task-specific limitation in real-world applications with diverse tasks.**
> - Thank you for your feedback. The MetaICL dataset encompasses evaluations for a wide range of tasks unseen during training, including reading comprehension, question answering, and natural language inference. In our paper, we demonstrate the strong task-generalization performance achieved by our compression method on this multi-task dataset.
> - However, as you pointed out, we optimize the model for specific application scenarios in experiments. To demonstrate its utility in more general use cases, we trained a single compression model and evaluated its performance across various scenarios. Specifically, we leverage **MetaICL training tasks and the conversation dataset, SODA, as our training data**, and then conduct **evaluations on MetaICL test tasks, LaMP, and DailyDialog**.
> For detailed experiment results, please refer to the **general response above**. The results presented in the table indicate that our approach are able to generalize across diverse scenarios without the need for additional fine-tuning. We will add this analysis in the revision.
>
> **Q. Hybrid approach combining task-specific and task-agnostic components.**
> - Your idea is indeed insightful. Task-specific compression offers the advantage of having a larger compression space since it focuses on compressing information relevant to a specific task, in contrast to task-agnostic compression, which aims to compress information in a general context. While task-agnostic methods are valuable for compression in generic scenarios, achieving superior compression performance in specific applications may require task-specific adaptations.
> - From a technical view, there is potential for significant benefits by integrating both approaches. For instance, one could first apply task-agnostic compression and then follow task-specific compression, combining the strengths of both methods. Alternatively, using a task-agnostic compression module as an initialization and then fine-tuning it in a task-specific context may reduce fine-tuning costs. We will incorporating these discussions into our paper.
>
> **Q. Additional resources for training**
> - Thank you for your question. In our paper, we compared our method with Gisting, Compressive Transformer, and in the appendix, RMT and AutoCompressor. During experiments, all these methods underwent **the same process of fine-tuning** on each dataset, identical to our method for a fair comparison. We adopted the training protocol from the Gisting paper and adjusted the training steps according to the size of each dataset.
>
> - Our approach is memory-efficient as it only requires training the LoRA adapter for compression. As demonstrated in Appendix Table 15, our parallelized training technique provides a **7x faster training speed** compared to existing recurrent memory approaches, RMT and AutoCompressor. Additionally, as indicated in Table 9, we used datasets containing around 100k samples for each application. This dataset size is similar to the recently released instruction finetuning dataset, Alpaca, and is roughly 10% of the size of LmSys-Chat-1M. In contrast, AutoCompressor relies on a massive training dataset containing up to 15 billion tokens for compression, which would demand significant resources and training costs when developing individual models in industry. In this point of view, our approach has advantages in resource-constrained environments by its relatively lower memory and data requirements.
>
> **Q. Analysis on performance gap**
> - Firstly, the existence of a performance gap can be attributed to our compression experiment setup, which demands generalization abilities for new tasks (MetaICL) / users (LaMP) / context histories (DailyDialog). In fact, while the MetaICL train loss for full context LLaMA-7B is **0.59** and CCM-concat's train loss is **0.61**, the test set loss diverges to **0.76** (full context) and **0.92** (CCM-concat), indicating the presence of a generalization gap. On the other hand, the Compressive Transformer exhibits a similar level of train loss, but it shows a test loss of **1.21**, indicating a larger generalization gap.

---

> ### Author Response · Authors · 2023-11-19
> **Author response (2)**
>
> **Q. Explanation for effectiveness**
> - As explained at the bottom of page 7, in conversation tasks like DailyDialog, where each context history contains a wide range of contextual information, the merging approach exhibits decreased performance when compared to the concatenation approach. Conversely, for context histories consisting of mutually complementary information, such as in MetaICL and LaMP, the merging method demonstrates its effectiveness.
> - We further discuss the effectiveness of our approach in specific scenarios. As shown in Table 8, our method excels in the accuracy metric. When evaluating the **generation performance with RougeL metric** in MetaICL, we obtain the following results:
> | Metric | No-context | Full-context | Gisting | Gisting-online | Compressive | CCM-merge | CCM-concat |
> |-|-:|-:|-:|-:|-:|-:|-:|
> | RougeL | 12.3 | 61.4 | 47.7 | 37.9 | 47.9 | 48.3 | **54.7** |
> | Accuracy (%) | 51.7 | 70.8 | 66.9 | 57.7 | 67.8 | 69.6 | **70.0** |
>
> - From the table, our method delivers the most accurate generation performance compared to other baselines. However, there is a larger decrease in performance compared to the full context, whereas in the case of accuracy, the performance drop is less than 1%. When we examine the generated outputs with compressed context, we observe instances where synonyms (e.g., "Different" and "Dissimilar" in the medical_questions_pair task) or variations in letter casing (e.g., "Hate" and "hate" in the tweet_eval_hate task) exist, indicating semantic equivalence but differences in expression. Based on these results, we believe our method is particularly effective in scenarios that **prioritize preference or nuances**, rather than returning precise expressions. We will incorporate these discussions into the revised version.
>
> **Q. More sophisticated memory update mechanisms**
> - Based on your advice, we have formulated a more **general form of the memory update** equation as follows:
> $\text{Mem}(t) = (1 - a_{t-1}) \text{Mem}(t-1) + a_{t-1} h(t), \text{for}\ a_{t-1} \in [0,1].$
> In our paper, we proposed an update method based on the arithmetic average (AA) of the compressed state up to the present time, i.e., $a_{t-1}=1/t$. Another natural design choice can be an exponential moving average (EMA), where $a_{t-1}$ is set to a constant value. This strategy weigh higher importance on recent information, when compared to the arithmetic average. In below, we train a model with EMA and compare it's test performance to the original CCM-merge (AA).
> | Time step | 1 | 2 | 4 | 8 | 12 |
> |-|-:|-:|-:|-:|-:|
> | EMA | 7.49 | 7.06 | 6.79 | 6.49 | 6.38 |
> | AA | 7.47 | 7.06 | 6.87 | 6.54 | 6.34 |
>
> - The table above provides perplexity results of the arithmetic average and EMA with $a_{t}=0.5$, in DailyDialog with LLaMA-7B. S The results indicate that both methods yield similar performance. When forming the compression state $h(t)$, our method involves referencing the previous memory, $\text{Mem}(t-1)$. We believe this enable preservation of overall context, even with exponentially decreasing coefficient for past states by EMA. We also note that CCM-concat can be interpreted as dynamically inferring coefficients for individual hidden states $h(t)$ through the transformer attention mechanism.
>
> **Q. Lack of comparison to recurrent memory approaches.**
> - Thank you for your feedback pointing out the significance of recurrent memory approaches as a baseline. In Appendix D, Table 15, we conduct a comparison with recent recurrent memory approaches, RMT and AutoCompressor. Direct comparison with LinearTransformer is non-trivial due to its structural modifications in the Transformer operations, making its application to foundation models like LLaMA challenging. We acknowledge the importance of this previous work and will add discussion in the related work section.
> - From comparison to RMT and AutoCompressor, we observed that our parallelized training technique significantly **improves training speed by approximately 7x** (Appendix, Table 15), while also achieving superior performance at a similar compression level. We recognize the importance of this comparison, and we will incorporate it into the main body of our paper.
>
> Thank you once again for the valuable feedback. If you have any remained questions, please let us know.

---

> > ### Comment · Reviewer_JGpv · 2023-11-21
> > **Thanks for the response**
> >
> > Thank the authors for the detailed responses. My concerns have been addressed, and I will keep my current recommendation for acceptance.
> >
> > Reviewer

---

> > > ### Author Response · Authors · 2023-11-21
> > >
> > > Thank you for your confirmation!
> > >
> > > Best regards,
> > > Authors

---

### Author Response · Authors · 2023-11-19
**General Response: General compression model**

We provide experimental results on generality test of our approach.
- In detail, we train LLaMA-7B on the **mixture of MetaICL training set (61 tasks) and SODA conversation dataset**. We basically follow the training recipe in Table 9 of our paper, while we train a model for 4k steps. We train the baselines Gisting and Compressive Transformer by using the same dataset and training protocol. We use the <COMP> token length of 2 for CCM-concat and 8 for CCM-merge. Finally, we **test the model on MetaICL test set (26 unseen tasks), LaMP, and DailyDialog** at the maximum time step (Table 8).
| Dataset | metric | Full context | No context | Gisting-online | Compressive | CCM-merge | CCM-concat |
|---|---|---|---|---|---|---|---|
| MetaICL | Accuracy | 70.0 | 53.6 | 59.9 | 65.0 | 67.8 | **68.7** |
|  | context_length | 580 | 0 | 32 | 32 | 8 | 32 |
| LaMP | Accuracy | 76.4 | 37.0 | 67.6 | 58.4 | 71.4 | **75.2** |
|  | context_length | 755 | 0 | 32 | 32 | 8 | 32 |
| DailyDialog | Perplexity | 7.02 | 11.51 | 9.04 | 9.19 | 8.22 | **7.61** |
|  | context_length | 222 | 0 | 24 | 24 | 8 | 24 |

- From the results above, we demonstrate that our method can generalize to datasets and scenarios unseen during training. Specifically, CCM-concat **maintains the best compression performance** compared to other methods considered by a large margin. While the CCM-merge shows moderate performances on test settings, we observe the method have increased performance degradation by compression, when compared to the scenario-specific settings in our submission (e.g., the LaMP accuracy degradation by compression increased from 1.2% to 5.1%). However, the other compression baselines have larger performance gap by compression, demonstrating our approach achieves the best generalization performance among the baaselines.

---

### Author Response · Authors · 2023-11-19
**Author responses are uploaded**

Dear reviewers,

Thank you for your valuable effort and time in providing helpful feedback on our work. We sincerely appreciate your encouraging comments, including “The paper addresses an important open challenge” (Reviewer JGpv, Svka), “The method is efficient in both training and inference” (Reviewer JGpv, JvdE, Svka), “The evaluation is comprehensive and diverse” (Reviewer JGpv, JvdE), “An interesting method” ( Reviewer UgoG).

We have carefully considered all the points raised and provided rebuttals accordingly. If you have any further questions, please let us know. We will reflect all discussions in the revision and release the code for reproducibility.

Best regards,
The authors

---

### Comment · Area_Chair_RKTJ · 2023-11-20
**Please engage in reviewer-author discussions**

Reviewers - I encourage you to read the authors' response carefully and let the authors know whether their response has addressed your comments.

---

### Meta-Review · Area_Chair_RKTJ · 2023-12-06

**Metareview:**

This paper introduces a method for dynamically compressing contextual information during online inference with large language models (LLMs) by existing gisting approaches to multi-context settings. This allows the model to efficiently handle continuously expanding contexts, such as those encountered in long conversations. The contributions are as follows.
- The proposed method condenses the context into a compact representation dynamically updated during online inference, reducing memory usage and computation compared to using the full context.
- A parallel training strategy using masked attention to learn the context compression in a single forward pass.
- A conditional adapter applied only to compression tokens to avoid overfitting to inputs during training.

Experiments demonstrate promising compression rates with minimal performance trade-off compared to using full context. In general, the reviewers appreciate the efficiency and effectiveness, and confirmed that most of the concerns are addressed during rebuttal with the additional experiments of training on a mixture of MetaICL training set (61 tasks) and SODA conversation dataset for testing generalization to MetaICL test set, LaMP, and DailyDialog. There is a remaining doubt about whether the model can be generalized to handling compression tokens at all positions. The AC acknowledged the concern, but it doesn't appear to be problematic in the empirical results. Thus, the AC recommend accepting this work.

**Justification For Why Not Higher Score:**

The performance and good compression rate are likely results of limiting to specific domains and data types (online LM interactions). It's questionable whether it can sustain when considering much more diverse tasks.

**Justification For Why Not Lower Score:**

Good performance, novel approach, solid experiments.

---

### Decision · Program_Chairs · 2024-01-16

Accept (poster)